# DO I LOOK LIKE A "CAT.N.01" TO YOU?
# A TAXONOMY IMAGE GENERATION BENCHMARK

## ABSTRACT

This paper explores the feasibility of using text-to-image models in a zero-shot setup to generate images for taxonomy concepts. While text-based methods for taxonomy enrichment are well-established, the potential of the visual dimension remains unexplored. To address this, we propose a comprehensive benchmark for Taxonomy Image Generation that assesses models' abilities to understand taxonomy concepts and generate relevant, high-quality images. The benchmark includes common-sense and randomly sampled WordNet concepts, alongside the LLM generated predictions. The 12 models are evaluated using 9 novel taxonomy-related text-to-image metrics and human feedback. Moreover, we pioneer the use of pairwise evaluation with GPT-4 feedback for image generation. Experimental results show that the ranking of models differs significantly from standard T2I tasks. *Playground-v2* and *FLUX* consistently outperform across metrics and subsets and the retrieval-based approach performs poorly. These findings highlight the potential for automating the curation of structured data resources.

## 1 INTRODUCTION

In recent years, Large Language Models (LLMs) and Visual Language Models (VLMs) have demonstrated remarkable quality across a wide range of single- and cross-domain tasks Esfandiarpoor et al. (2024); Esfandiarpoor & Bach (2023); Du et al. (2023); Jiang et al. (2024b). Their capabilities also expand to the tasks traditionally dominated by human input, such as annotation and data collection Tan et al. (2024). At the very same time, the urge for manually created datasets and databases still remains popular, as more accurate and reliable Zhou et al. (2023), even though they are time-consuming and expensive to be kept up-to-date.

In this paper, we focus on taxonomies — lexical databases that organize words into a hierarchical structure of "IS-A" relationships. WordNet Miller (1998) is the most popular taxonomy for English, forming the graph backbone for many downstream tasks Mao et al. (2018); Lenz & Bergmann (2023); Fedorova et al. (2024). In addition to textual data, taxonomies also extend to visual sources, e.g. ImageNet Deng et al. (2009). ImageNet is built upon the WordNet taxonomy by associating concepts or "synsets" (sets of synonyms, aka lemmas) with thousands of manually curated images. However, it covers a very small portion of WordNet taxonomy (5,247 out of 80,000 synsets in total, 6.5%).

From the visual perspective, Text-to-Image models are widely used for the visualizations Ng et al. (2024); Sha et al. (2023), but only occasionally for taxonomies Patel et al. (2024a). Therefore, there is limited knowledge about how well text-to-image models are capable of visualizing concepts of different level of abstraction in comparison to humans Liao et al. (2024). Image generation for taxonomies could be quite specific and require additional research: Figure 1 highlights the key differences in prompt usage for the DiffusionDB dataset Wang et al. (2023) and WordNet-3.0. Moreover, the output taxonomy-linked depictions should aim succinctly portraying the synset's core idea and/or sometimes revealing insights about the concept that are challenging to convey textually.

Therefore, in this paper, we address this gap by investigating the use of automated methods for updating taxonomies in the image dimension (depicting). Specifically, we develop an evaluation benchmark comprising 9 metrics for Taxonomy Image generation using both human and automatic evaluation and a Bradley-Terry model ranking in line with recent top-rated evaluation methodology Chiang et al. (2024a); Zheng et al. (2023b). Suprisingly, our task yields different rankings for models

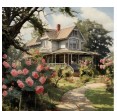 Suburban landscape, trees and roses, cut house with a wraparound porch
*(from DiffusionDB Large)*

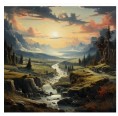 landscape
*(from WordNet 3.0)*

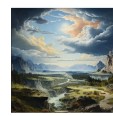 landscape
(an extensive mental viewpoint)
*(from WordNet 3.0, w/ definition)*

Figure 1: Comparison of generations of the *Playground* model for the input prompt from the DiffusionDB dataset and available inputs from the WordNet-3.0. It can be seen, that the input from the TTI dataset is more detailed and the inner model representation could be misguiding even when the difinition is given.

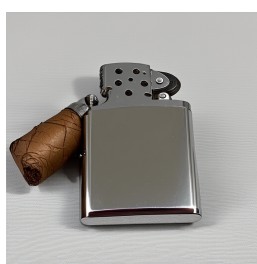 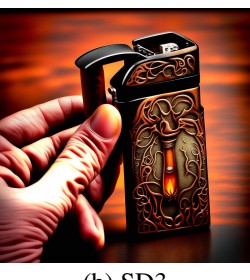 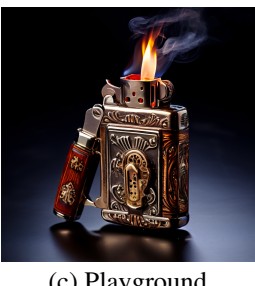 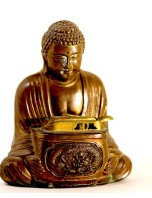

| (a) Kandinsky 3 | (b) SD3 | (c) Playground | (d) Retrieval |

Figure 2: The example of a generation and retrieval results for *cigar lighter*. As can be observed, the generation approach is significantly superior to the retrieval approach, as the retrieved image is quite unconventional.

compared to those in text-to-image benchmarks Jiang et al. (2024a), higlighting the task importance. We also uncover that modern Text-to-Image models outperform traditional retrieval-based methods in covering a broader range of concepts, highlighting their ability to better represent and visualize these previously underexplored areas.

The contributions of the paper are as follows:

- We propose a benchmark comprising 9 metrics, including
  - several taxonomy-specific text-to-image metrics (grounded with theoretical justification drawing on KL Divergence and Mutual Information)
  - pairwise preference evaluation with GPT-4 for text-to-image generation and analyze its alignment with human preferences, biases, and overall performance.
- We test on the dataset specifically designed for Taxonomy Image Generation task, which presents challenges that were previously unaddressed in text-to-image research.
- We are the first to evaluate the performance of the 12 publicly available Text-to-Image models to generate images for WordNet concepts on the developed benchmark.
- We publish the dataset of the images generated by the best Text-to-Image approach from the benchmark that fully covers WordNet-3.0 extending the ImageNet dataset.

## 2 DATASETS

This section provides an overview of the datasets used to evaluate the performance of text-to-image (TTI) models. It includes the Easy Concepts dataset, the TaxoLLaMA test set derived from WordNet, and the predictions generated by the TaxoLLaMA model. The aim of the datasets is to assess the models' sensitivity to easier/harder dataset and to existing/AI-generated entities.

### 2.1 EASY CONCEPT DATASET

The Easy Concepts dataset from Nikishina et al. (2023) comprises 22 synsets selected by the authors as common-sense concepts (e.g. *"coin.n.01, chromatic_color.n.01, makeup.n.01, furniture.n.01"*, etc.). We extend this list by including their direct hyponyms ("children nodes"), following the

methodology outlined in the original paper and based on the English WordNet (Miller, 1998). The resulting dataset comprises 483 entities and represents a broader set of common knowledge entities.

## 2.2 RANDOM SPLIT FROM WORDNET

To generate the second dataset, we use the algorithm from TaxoLLaMA (Moskvoretskii et al., 2024b). We randomly sample the nodes the following types of hierarchical relations between synsets:

- **Hyponymy (Hypo)**: from a broader word (*"working_dog.n.01"*) to a more specific (*"husky.n.01"*). Here we take a broader word for image generation.
- **Hypernymy (Hyper)**: from a more specific word (*"capuccino.n.01"*) to a broader concept (*"coffee.n.01"*). Here we take a more specific word for image generation.
- **Synset Mixing (Mix)**: nodes created by mixing at least two nodes (e.g. *"milk.n.01"* is a *"beverage.n.01"* and a *"diary_product.n.01"*). Here we take the node created by mixing for depiction.

The algorithm for sampling uses a $0.1$ probability for sampling Hyponymy, a $0.1$ probability for sampling Synset Mixing, and a $0.8$ probability for sampling Hypernymy. The dominance of Hypernymy is necessary because it is the most useful relation for training TaxoLLaMA Moskvoretskii et al. (2024a). To mitigate this bias, the probabilities of occurrence in the test set differ between cases: for Hypernymy is set very low at $1 \times 10^{-5}$, higher for Hyponymy at $0.05$, and highest for Synset Mixing at $0.1$, as these cases are rare.

The resulting test set includes 1,202 nodes: 828 from Hypernymy relations, 170 from Synset Mixing relations, and 204 from Hyponymy relations.

## 2.3 LLM PREDICTIONS DATASETS

As our final goal is depicting of the new concepts for taxonomy extension, we should also test TTI models with LLM predictions rather than ground-truth synsets. Therefore, we finetune an LLM model on the Taxonomy Enrichment task to use its predictions and assess the sensitivity of text-to-image (TTI) models to AI-generated content.

The workflow comprises three steps: (i) exclude the Easy Concept dataset and random split from the overall WordNet data for LLM model training; (ii) train the updated version of TaxoLLaMA with LLaMA-instruct-3.1 Dubey et al. (2024); (iii) solve the Taxonomy Enrichment task for the test data to generate concepts for vizualization. When training the TaxoLLaMA-3.1 model, we follow the methodology outlined in Moskvoretskii et al. (2024b).

This process resulted in 1,685 items. To match the original WordNet synsets, we generate definitions for every generated node with GPT4, described in Appendix C.

## 3 MODELS

In this section, we describe ten TTI models and one Retrieval model (12 in total) and the details of image collection. Table 1 comprises the full list of the models compared in the evaluation benchmark, their description can be found in Appendix B.

An example of a prompt for image generation is demonstrated below, details are described in Appendix F. We perform experiments with two versions of the prompt: with and without definition. It is worth noting that adding definitions does not turn the task into "standard instruction following." In TTI settings, definitions are not a typical or natural form of instruction, therefore it is an additional informative diagnostic, revealing how models retrieve fine-grained taxonomic meaning. Figure 2 shows the example of a generation and retrieval results guided by the prompt:

```
TEMPLATE: An image of <CONCEPT> (<DEFINITION>)

EXAMPLE: An image of cigar lighter (a lighter for cigars or
cigarettes)
```

```
Please act as an impartial judge and evaluate the quality of the images provided by two AI
image assistants to the user prompt displayed below.

You should choose the assistant that provides an image that follows the user's instructions
better and reflects the user's prompt main concept better.
Your evaluation should consider factors such as image-text alignment, relevance, accuracy,
depth, and fidelity (overall image quality).
Begin your evaluation by comparing the two images and provide a short explanation.

Avoid any position biases and ensure that the order in which the
responses were presented do not influence your decision.

Do not allow the size of the images to influence your evaluation.
Do not favor certain names of the assistants.
Be as objective as possible.

After providing your explanation, output your final verdict by strictly following this format:

"[[A]]" if assistant A is better, "[[B]]" if assistant B is better, "[[C]]" for a tie and "
[[D]]" if both images are bad.

[User Prompt]

inherited disorder (a genetic condition passed down from parents to their offspring,
oftenresulting in physical or mental health problems))
```

Figure 3: LLM prompt example for evaluating text-to-image assistants.

| Model name | Size | Model Family | Paper |
|---|---|---|---|
| SD-v1-5 | 400M | | Rombach et al. (2022) |
| SDXL | 6.6B | | Podell et al. (2024) |
| SDXL Turbo | 3.5B | U-Net | Liu et al. (2024) |
| Kandinsky 3 | 12B | | Arkhipkin et al. (2023) |
| Playground-v2-aesthetic | 2.6B | | Li et al. (2023) |
| Openjourney | 123M | | Prompthero (2023) |
| IF | 4.3B | | DeepFloyd.Lab (2023) |
| SD3 | 2B | Diffusion | Esser et al. (2024) |
| PixArt-Sigma | 900M | Transformers | Chen et al. (2024b) |
| Hunyuan-DiT | 1.5B | | Li et al. (2024) |
| FLUX | 12B | | BlackForestLabs (2024) |
| Wikimedia Commons[1] | - | Retrieval | Ferrada et al. (2018) Jones & Oyen (2022) |

Table 1: List of the approaches evaluated in the Taxonomy Image Generation benchmark.

## 4 EVALUATION

In this section, we describe the evaluation process and metrics.

Our evaluation consists of 9 metrics that we assess to provide a comprehensive evaluation using the latest methods. To formally define our metrics, let $V$ be a finite set of concepts $v$, $A(v) \subseteq V$ the set of hypernyms for $v$, and $N(v) \subseteq V$ the set of cohyponyms for $v$. Let $X^j$ be the set of all possible images $x^j$ in a finite model space $j \in J$ and $|J| = 12$ in our case. We define a mapping $g^j : V \to X^j$ for each model $j$, which assigns an image to each concept $v$.

### 4.1 PREFERENCES METRICS

**ELO Scores** We evaluate the ELO scores of the model by first assigning pairwise preferences, similar to the modern evaluation of text models Chiang et al. (2024a). Each object $v$ is assigned two uniformly sampled random models $A, B \sim \mathbb{U}[J]$, and their outputs $(x^A, x^B)$ engage in a battle. Then, either a human assessor or GPT-4 serves as a function to assign a win to model A (0) or model

| Metric | Mean | Ground Truth | | | | Predicted | | | |
|---|---|---|---|---|---|---|---|---|---|
| | | Easy | Hypo | Hyper | Mix | P-Easy | P-Hypo | P-Hyper | P-Mix |
| ELO GPT (w/ def) | Playground | Playground | Playground | Playground | Playground | PixArt | PixArt | Playground | SD3* |
| ELO GPT (w/o def) | Playground | Kandinsky3 | PixArt* | Playground | FLUX | FLUX | Playground | Playground | Kandinsky |
| ELO Human (w def) | FLUX | Playground / DeepFloyd | Kandinsky3 | FLUX | Playground | FLUX | FLUX | Playground | PixArt |
| ELO Human (w/o def) | FLUX | SD3 | Kandinsky3 | Playground | PixArt | FLUX | FLUX | SDXL | SDXL |
| Reward Model (w/ def) | Playground | Playground | Playground | Playground | Playground | Playground | Playground | Playground | Playground |
| Reward Model (w/o def) | Playground | Playground | Playground | Playground | Playground | Playground | Playground | Playground | Playground |
| Lemma Similarity | SDXL-turbo | SDXL-turbo | SDXL-turbo | SDXL-turbo | SDXL-turbo | SDXL-turbo | SDXL-turbo | SDXL-turbo | SDXL-turbo |
| Hypernym Similarity | SDXL-turbo | FLUX | FLUX / SDXL-turbo | SDXL-turbo | FLUX / SDXL-turbo | SDXL-turbo | SDXL-turbo | Draw | Draw |
| Cohyponyms Similarity | SDXL-turbo | FLUX | SDXL-turbo | SDXL-turbo | SDXL-turbo | SDXL-turbo | SDXL-turbo | SDXL-turbo | SDXL-turbo / SDXL |
| Specificity | SD1.5 | SD1.5 / Playground | SD1.5 | SD1.5 / Playground | Draw | Draw | SDXL-turbo / SDXL | SDXL-turbo | SDXL-turbo |
| FID | SD1.5 | FLUX | FLUX | FLUX | HDiT | FLUX | FLUX | FLUX | DeepFloyd |
| IS | SD3 | PixArt | Playground | Playground | SD3 | SD3 | FLUX | FLUX / Retrieval | Playground |

Table 2: Summary of the Top-1 model for each metric and subset. Each cell shows the best-rated model. If two models tie, both are listed with a slash; if more than two tie, "Draw" is written, indicating insufficient specificity. Results marked with * have negligible differences within the confidence interval. Subsets and models are described in Sections 2 and 3.

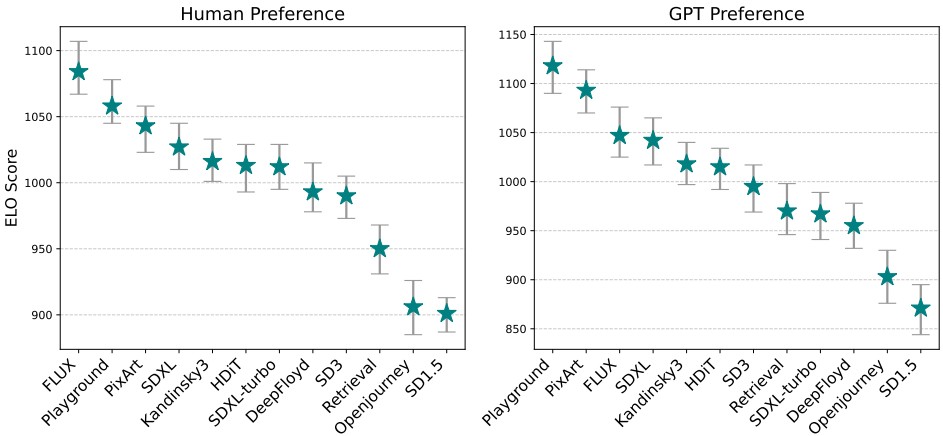

Figure 4: ELO scores for human and GPT4 preferences. The prompt includes the definition. Overall Spearman correlation of model rankings remains significantly high at $0.92$, $p$-value $\leq 0.05$.

B (1), represented as $f(v, x^A, x^B) \in \{0, 1\}$. Ties are omitted in both the notation and the BT model. To compute ELO scores, the likelihood is maximized with respect to the BT coefficients for each model, which correspond to their ELO scores. More details of the approach can be found in the Chatbot Arena paper Chiang et al. (2024a).

Following this methodology, we calculate the ELO score based on the Bradley-Terry (BT) model with bootstrapping to build 95% confidence intervals. The Bradley-Terry model Bradley & Terry (1952) is a probabilistic framework used to predict the outcome of pairwise comparisons between items or entities. It assigns a latent strength parameter $\pi$ to each item $i$ and the probability that item $i$ is preferred over item $j$ is given by: $P(i > j) = \frac{\pi_i}{\pi_i + \pi_j}$. Here, $\pi_i, \pi_j > 0$ represent the strengths of items $i$ and $j$, respectively. The parameters are typically estimated from observed comparison data using maximum likelihood estimation. We also adopt a labeling technique that includes the "Tie" and "Both Bad" categories, indicating cases where the models are equally good or both produce poor outputs. We modify the prompt from previous studies evaluating text assistants Zheng et al. (2023a) for images, as presented in Figure 3 (also see prompt 7 in Appendix E for more details).

GPT-4 is only one of the nine metrics we report, and it is used as a single comparative signal, not as the core evaluation mechanism. Therefore, we also conduct the **Human ELO Evaluation** along with the **GPT-4 ELO Evaluation** on 3370 pair images from two different models ($\approx 600$ samples from each model). For Human ELO, we employ 4 assessors expert in computational linguistics, both male and female with at least bachelor degrees. The Spearman correlation between annotators is 0.8 ($p$-value $\leq 0.05$) for the images generated with definitions. For the automatic calculation of the ELO score we use GPT-4, which is highly correlated with human evaluations Zheng et al. (2023a) and has proven to be an effective image evaluator on its own Cui et al. (2024), as well as a great pairwise preferences evaluator Chen et al. (2024a).

**Reward Model** We utilize the reward model from a recent study Xu et al. (2024), which is trained to align with human feedback preferences, focusing on text-image alignment and image fidelity.

This score demonstrates a strong correlation with human annotations and outperformed the CLIP Score and BLIP Score. Formally, this metric is similar to ELO Scores, as the reward model was tuned using preferences and the BT model. However, the key difference is that each object $v$ is assigned a real-valued score and takes only one model image $x^j$ as input: $f_{reward}(v, x^j) \in \mathbb{R}$.

## 4.2 SIMILARITIES

In this section, we introduce novel similarity metrics that leverage taxonomy structure and are derived from KL Divergence and Mutual Information, with formal probabilistic definitions provided in Appendix D. They all have CLIP similarities under the hood, which have been already validated against human judgements Hessel et al. (2021). This ensures that our metrics, by extension, are aligned with human judgements.

In practice, we approximate the probabilities using CLIP similarity Hessel et al. (2021), as it is the most reliable measure of text-image co-occurrence.

Formally, CLIP model $C(\text{text or image}) \in \mathbb{R}^{hidden\_dim}$, we calculate the cosine similarity between the embedding of concept $v$ and the embedding of image $x^j$, resulting in the score $\text{sim}(C(v), C(x^j))$

**Lemma Similarity** reflects how well the image aligns with the lemma's textual description; is defined as

$$S_{\text{lemma}}(v,x) := P(X=x|v) \approx \text{sim}(C(v), C(x^j)). \tag{1}$$

**Hypernym Similarity** reflects how similar the image is on average to the lemma hypernyms; is defined as

$$S_{\text{hyper}}(v,x) := P(X=x|A(v)) = \frac{1}{|A(v)|} \sum_{a \in A(v)} P(X=x|a) \approx \frac{1}{|A(v)|} \sum_{a \in A(v)} \text{sim}(C(a), C(x)). \tag{2}$$

**Cohyponym Similarity** measures how similar the image is, on average, to the cohyponyms; is defined as

$$S_{\text{cohyponym}}(v,x) := P(X=x|N(v)) = \frac{1}{|N(v)|} \sum_{n \in N(v)} P(X=x|n) \approx \frac{1}{|N(v)|} \sum_{n \in N(v)} \text{sim}(C(n), C(x)). \tag{3}$$

This metric should be interpreted in conjunction with Specificity, as a high Cohyponym Score paired with low Specificity does not necessarily indicate good generation.

In the T2I domain, it is not feasible to define "accuracy" in the traditional sense. It is difficult to determine whether the reflection of a concept is entirely correct or completely incorrect. This challenge is inherent to the nature of T2I tasks and is shared by other studies in this domain. To address this limitation, we propose an analogous measure to assess how well the image reflects the concept. We use the probability of the concept with respect to the generated image, denoted as $P(X = x|v)$, which is derived from Lemma Similarity. To further refine this measure, we also consider how well the generated image fits into the surrounding conceptual space by evaluating Hypernym Similarity and Cohyponym Similarity. These additional metrics help capture how accurately the image represents the broader context of the concept.

In order to understand how well hypernym and co-hyponym similarities correlate with human semantic understanding, we calculated Spearman correlation of the model ranks assigned by these metrics and ranks assigned through the human evaluation ($\rho \approx 0.911, p \leq 0.00004$ for Hypernym CLIP-Score, $\rho \approx 0.871, p \leq 0.00022$ for Co-hyponym CLIP-Score). This demonstrates that the proposed metrics capture relations that humans reliably recognize. Additionally, we would like to specify that WordNet is a human-curated taxonomic structure, which already embeds expert semantic judgments about hypernymy, hyponymy, and siblinghood. WordNet is itself the product of extensive manual linguistic annotation, and therefore provides a reliable human-grounded semantic foundation. Therefore, our evaluation benchmark already assesses how well models connect these human-defined concepts to visual evidence.

**Specificity** helps to ensure that the image accurately represents the lemma rather than its cohyponyms with the relation of the CLIP-Score to the Cohyponym CLIP-Score $\frac{S_{\text{hyper}}(v,x)}{S_{\text{cohyponym}}(v,x)}$

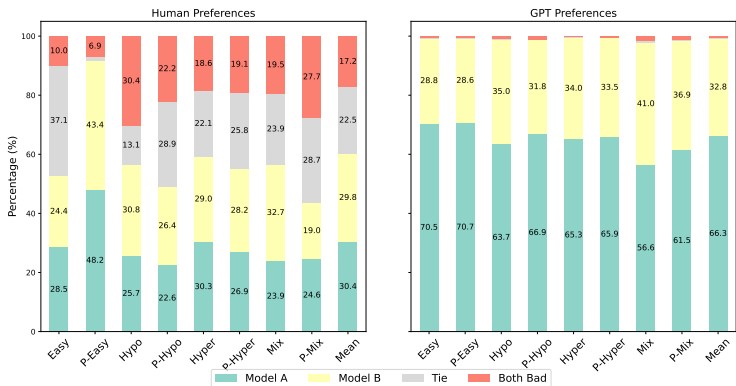

Figure 5: Distribution of preferences for Human and GPT across subsets in percentage (with definition).

This metric generalizes the In-Subtree Probability, as proposed in Baryshnikov & Ryabinin (2023). The key advantage of our metric is that it does not depend on a specific ImageNet classifier and can be applied to any type of taxonomy node.

### 4.3 FID AND IS

We evaluate the Inception Score (IS) Salimans et al. (2016) and the Fréchet Inception Distance (FID) Heusel et al. (2017). IS is primarily used to assess diversity, while FID measures image quality relative to true image distributions. In our case, we calculate FID based on retrieved images, meaning that in this specific setting, FID reflects the "realness" or closeness to retrieval rather than the semantic correctness of an image.

### 5 RESULTS & ANALYSIS

The summary of the main results are presented in Table 2 and in Appendix H: they show the best model for each subset and each metric. Additionally, we provide an error analysis in Appendix I and discuss the strengths and weaknesses of the best-performing models.

**ELO Scores** The preferences of human evaluators and GPT-4 resulted in the ELO Scores are shown in Figure 4. FLUX and Playground rank the first and the second across both GPT-4 and human assessors, with PixArt securing the third place. While the other rankings are less consistent—likely due to the difficulty in distinguishing between middle-performing models—the overall Spearman correlation of model rankings remains significantly high at $0.88$, $p$-value $\leq 0.05$.

Ranking without definitions is presented in Figure 8 in Appendix G, where FLUX ranks first for the Human preferences and Playground for the GPT Preference. However, the confidence intervals for the GPT Preference suggest it is not a definitive winner, as it ranks similarly to PixArt. The correlation between human and GPT-4 rankings is $0.73$, $p \leq 0.05$, which, while lower, is still strong.

At the same time, we found no correlation between raw scores for individual battles. This issue stems from a strong bias toward the first option, as illustrated in Figure 5 and the Confusion Matrix in Figure 12 in Appendix G, a bias not exhibited by humans. Most TTI models benefit from definitions in their input which exposes high human-GPT alignment, as shown in Figure 6 in Appendix C.

**Reward Model** The results from the Reward Model, introduced in a previous study Xu et al. (2024), show that Playground is the most preferred model, followed by PixArt and FLUX, with no significant differences between the latter, as shown in Figure 10 in Appendix G. Overall, the Reward Model demonstrates a high correlation with human evaluations (0.79) and a moderate correlation with GPT-4 (0.59). Playground is also the preferred model across all subsets, as illustrated in Fig-

ure 11 in Appendix G, while Figure 16 in Appendix H highlighting the statistical significance of these comparisons.

**Similarities** for lemmas, hypernyms, and cohyponyms consistently shows the dominance of SDXL-turbo across all subsets and FLUX for Easy Ground Truth subset. This result differs from AI preferences, possibly due to CLIP-Score focusing solely on text-image alignment without accounting for image quality. It is also noteworthy that SDXL-turbo ranks higher than SDXL, despite being a distilled version of the latter. The distillation process may have preserved more of the image-text alignment features while reducing overall image quality, as suggested in the original paper, while other models are not distilled or are specifically tuned to match user preferences.

**Specificity** shows no clear dominance, although the top models are SDXL-turbo, SD1.5, and Playground. SD1.5 ranks first in several subsets, though it performs poorly in terms of user preferences. Moreover, this result indicates that Playground's generations can be specific to the precise lemma, aligning both with preference and specificity.

**FID** results, presented in Table 8 and Table 10 in Appendix H, demonstrate that on average SD1.5 performs best, however FLUX dominates across nearly all subsets. We associate this performance with a stronger focus on reconstructing open-source crawled images, rather than aligning with human preferences and text-image alignment, however FLUX balancing to also appeal to human judgments.

**IS** results in Table 8 and Table 9 in Appendix H indicate that SD3, Playground, and Retrieval rank first across different subsets, suggesting their generations are perceived as "sharper" and more "distinct". All versions of SDXL and SD3 do not benefit from the definitions, likely due to the specific characteristics of the SD family.

**Overall** Our results show that Playground and FLUX are among the top models across different metrics, both with and without definitions. While PixArt also demonstrates strong results, it is preferred by AI evaluations more than human preferences, indicating that the preference may be more AI-Judge specific. However, the results are more heterogeneous for specificity, which measures how well the model reflects the concept itself and not its neighbors. Models from the SD family perform differently on different metrics and subsets, indicating that even when models trained with CLIP alignment may not guarantee specificity to the precise concept and the ability to reflect more detailed information of the node.

## 6 RELATED WORK

In this section, we describe existing evaluation benchmarks for both texts and images and provide an overview of text-to-image generation models. We do not provide an overview on the existing taxonomy-related tasks and approaches and refer to Zeng et al. (2024) and Moskvoretskii et al. (2024b).

**Evaluation Benchmarks** Popular benchmarks for language models include GLUE Wang et al. (2019) and SuperGLUE Sarlin et al. (2020), MTEB Muennighoff et al. (2023), SQuAD Rajpurkar et al. (2016), MT-Bench Zheng et al. (2023c) and others. For Text2Image Generation, there are benchmarks such as MS-COCO Lin et al. (2014), Fashion-Gen Rostamzadeh et al. (2018) or ConceptBed Patel et al. (2024b). There are also platforms for interactive comparison of AI models, based on ELO-rating: LMSYS Chatbot Arena Chiang et al. (2024b) for LLM and GenAI Arena Jiang et al. (2024a) for comparing text-to-image models. Moreover, due to latest AI's abilities, "LLM-as-a-judge" evaluation emerged: text or image generation outputs of different models are compared by another model, see Zheng et al. (2023c); Wei et al. (2024); Chen et al. (2024a).

**Text-to-Image Generation Models For Taxonomies** Image generation has recently received significant attention in the field of machine learning. Previously, Generative Adversarial Networks (GANs) (Goodfellow et al., 2014) and Variational Autoencoders (VAEs) (Kingma & Welling, 2014)

were primarily used for this purpose. However, diffusion-based methods have now become the dominant approach and are widely used for the visualizations Ng et al. (2024); Sha et al. (2023), but only occasionally for taxonomies Patel et al. (2024a).

To the best of our knowledge, the existing work on the evaluation of images for taxonomies comprises the paper of Baryshnikov & Ryabinin (2023), which introduces In-Subtree Probability (ISP) and Subtree Coverage Score (SCS), which are revisited in our paper. Recently, Liao et al. (2024) introduced a novel task of text-to-image generation for abstract concepts. The benchmark from Patel et al. (2024a) addresses grounded quantitative evaluations of text conditioned concept learners and the Zhang et al. (2024) also operates the notion of concepts for images when developing the concept forgetting and correction method.

## 7 CONCLUSION

We have proposed the Taxonomy Image Generation benchmark as a tool for the further evaluation of text-to-image models in taxonomies, as well as for generating images in existing and potentially automatically enriched taxonomies. It consists of 9 metrics and evaluates the ability of 12 open-source text-to-image models to generate images for taxonomy concepts. Our evaluation results show that Playground Li et al. (2023) ranks first in all preference-based evaluations.

## ETHICAL CONSIDERATIONS

In our benchmark, we utilized several text-to-image models as well as text assistants for evaluating and generating new concepts. While these models are highly effective for creating creative and novel content, both textual and visual, they may exhibit various forms of bias. The models tested in this benchmark have the potential to generate malicious or offensive content. However, we are not the creators of these models and focus solely on evaluating their capabilities; therefore, the responsibility for any unfair or malicious usage lies with the users and the models' authors.

Additionally, our fine-tuning of the LLaMA 3.1 model was conducted using safe prompts sourced from WordNet. Although applying quantization and further fine-tuning could potentially reduce the model's safety, we did not observe any unsafe or offensive behavior during our testing.

## REPRODUCIBILITY CONSIDERATIONS

We publish all datasets, generated wordnet images and collected preferences in an anonymous repo.

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

## A  LIMITATIONS

- Our evaluation focuses on open-source text-to-image models, as they are more convenient and cost-effective to use in any system than models relying on an API. Additionally, open-source models offer the flexibility for fine-tuning, which is not possible with closed-source models. However, it would be valuable to explore how closed-source models perform on this task, as our benchmark depends solely on the quality of the generated images.

- Preferences using GPT-4 were obtained with the use of Chain of Thought reasoning, following previous studies to optimize the prompt Zheng et al. (2023a). However, we did not utilize multiple generations with a majority vote to improve consistency, nor did we rename the models, which could help reduce positional bias. Additionally, we did not perform multiple runs with models alternating positions, as each model could appear in position A or B with equal probability. The Bradley-Terry model of preferences compensates for such inconsistencies and provides robust scoring, given a sufficient number of preference labels, as noted in a previous study Zheng et al. (2023a). Our assumption is further supported by the high correlation in the resulting rankings, even though the correlation between raw preferences is close to zero.

- Metrics based on CLIP-Score may be biased toward the CLIP model and lack specificity if CLIP is unfamiliar with or unspecific to the precise WordNet concept. Additionally, models could be fine-tuned to optimize for this particular metric. To mitigate this bias, we propose incorporating preferences from AI feedback and also employ preferences from human feedback to provide a more balanced and comprehensive evaluation.

- The Inception Score relies on the InceptionV3 model, which is specific to ImageNet1k. We included this metric as it is traditionally used to measure overall text-to-image performance. However, to address this potential bias, we introduced CLIP based metrics as well as generalization of the ISP metric from a previous study Baryshnikov & Ryabinin (2023), which also relies on an ImageNet1k classifier, but we supplemented it with the use of CLIP to provide a broader evaluation.

- The benchmark focuses mainly on WordNet concepts, which may limit generalization to other (multilingual) taxonomies or domains that differ in structure. Extending the benchmark to other resources (e.g. Wikidata, ConceptNet) would require substantial additional design and alignment work and is therefore a separate research direction. In the present paper, we intentionally focus on WordNet because it provides the clearest foundation for linguistic evaluation of model representations.

## B  TTI MODELS DESCRIPTION

To generate the images, we employed ten models and one retrieval approach. It results in 12 systems in total.

### B.1  U-NET-BASED MODELS

Models based on the architecture:

- **SD-v1-5** (400M) (Rombach et al., 2022) is a SD-v1-2 fine-tuned on 595k steps at resolution 512x512 on "laion-aesthetics v2 5+" and 10% dropping of the text-conditioning to improve classifier-free guidance sampling.

- **SDXL** (6.6B) (Podell et al., 2024). The U-Net within is 3 times larger comparing to classical SD models. Moreover, additional CLIP (Radford et al., 2021) text encoder is utilized increasing the number of parameters.

- **SDXL Turbo** (3.5B) (Liu et al., 2024) is a distilled version of SDXL-1.0.

- **Kandinsky 3** (12B) (Arkhipkin et al., 2023). The sizes of U-Net and text encoders were significantly increased in comparison to the second generation.

- **Playground-v2-aesthetic** (2.6B) (Li et al., 2023) has the same architecture as SDXL, and is trained on a dataset from Midjourney[2].

---

[2] https://www.midjourney.com

- **Openjourney** (123M) (Prompthero, 2023) is also trained on Midjourney images.

## B.2 DIFFUSION TRANSFORMERS MODELS

Diffusion Transformers (DiTs) models:

- **IF** (4.3B) (DeepFloyd.Lab, 2023). A modular system consisting of a frozen text encoder and three sequential pixel diffusion modules.
- **SD3** (2B) (Esser et al., 2024) is a Multimodal DiT (MMDiT). The authors used two CLIP encoders and T5 (Raffel et al., 2020) for combining visual and textual inputs.
- **PixArt-Sigma** (900M) (Chen et al., 2024b). The authors employed novel attention mechanism for the sake of efficiency and high-quality training data for 4K images.
- **Hunyuan-DiT** (1.5B) (Li et al., 2024) is a text-to-image diffusion transformer designed for fine-grained understanding of both English and Chinese, using a custom-built transformer structure and text encoder.
- **FLUX** (12B) BlackForestLabs (2024) is a rectified flow Transformer capable of generating images from text descriptions. It is based on a hybrid architecture of multimodal and parallel diffusion transformer blocks.

## B.3 RETRIEVAL

We retrieved images from Wikimedia Commons[3], following previous studies Ferrada et al. (2018); Jones & Oyen (2022). For 3370 total items, this process resulted in 1,790 unique images. For 20 concepts (32 dataset entities), no images were found. For 146 lemmas, the search returned images that had already been retrieved, likely due to the similarity of the concepts searched. We use the top-1 output from the main image search engine[4].

## C DEFINITIONS ANALYSIS

We also analyzed how different models benefit from the inclusion of definitions in the TTI prompt, examining the change in winning battles with definitions (all models are provided with definitions), as depicted in Figure 6. Most models benefit from definitions according to human evaluation, though the trend is milder in GPT-4 evaluations, with preferences for Kandinsky3 and SD1.5 even dropping significantly. Despite the outlier of Kandinsky3, the overall trend between GPT-4 and human evaluations highlights the alignment of GPT-4's judgments with human preferences.

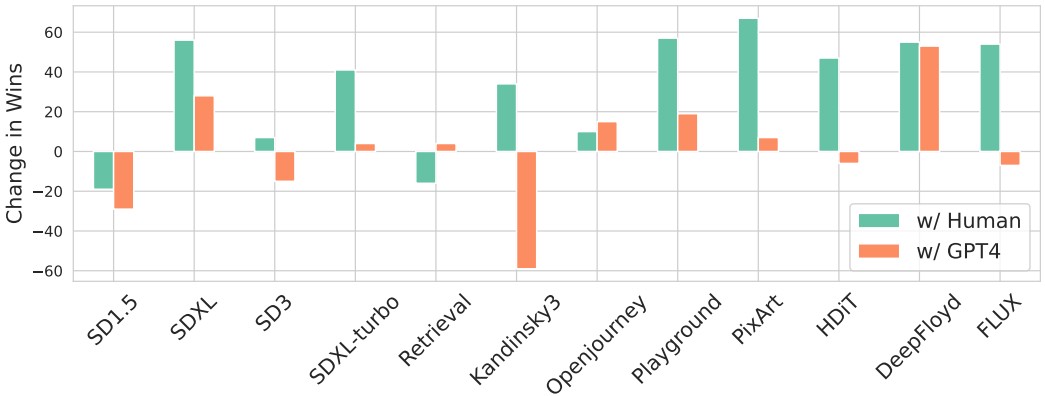

Figure 6: Summary change in battle wins with added definition in prompt.

---

[3] https://commons.wikimedia.org/

[4] For the lemma "coin", the search URL is https://commons.wikimedia.org/w/index.php?search=coin&title=Special:MediaSearch&go=Go&type=image

# D  METRICS DEFINITION

Let $V$ be a finite set of concepts (lemmas), where each $v \in V$ represents a semantic category. Let $(\mathcal{X}, \mathcal{A}, \mu)$ be a measurable space representing the image domain, where $\mathcal{X}$ is the set of all possible images, $\mathcal{A}$ is a $\sigma$-algebra of measurable subsets of $\mathcal{X}$, and $\mu$ is a base measure.

For each concept $v \in V$, we assume a probability measure $P(\cdot \mid v)$ on $(\mathcal{X}, \mathcal{A})$ such that $P(X \in A \mid v)$ represents the probability that an image $X$ generated under the concept $v$ lies in the measurable set $A$. In other words, each concept $v$ defines a probability distribution over the image space $\mathcal{X}$.

## D.1  LEMMA SIMILARITY

**Definition** (Lemma Similarity). *Given a concept $v \in V$ and an image $x \in \mathcal{X}$, the* Lemma Similarity *is defined as:*

$$S_{lemma}(v, x) := P(X = x \mid v).$$

This measures the likelihood of observing the image $x$ under the assumption that the concept $v$ is the generating source. According to Theorem 1, it also reflects the likelihood of the concept $v$ given the image $x$, offering a principled way to assess how well an image aligns with the semantic properties of a concept. Maximizing this metric implies a stronger alignment between the concept and the generated image.

**Theorem 1.** *Let $V$ be a finite set of concepts, and suppose the prior distribution is uniform $P(V = v) = \frac{1}{|V|} \; \forall v \in V$. Then, $\forall x \in X \; \forall v \in V \; \arg\max_{i \in V} S_{lemma}(v, x) \propto \arg\max_{i \in V} P(V = v \mid X = x)$.*

*Proof.* By Bayes' rule, the posterior probability of concept $i$ given an image $x$ is:

$$P(V = i \mid X = x) = \frac{P(X = x \mid i)P(V = i)}{\sum_{v \in V} P(X = x \mid v)P(V = v)} = \frac{P(X = x \mid i) \cdot \frac{1}{|V|}}{\sum_{v \in V} P(X = x \mid v) \cdot \frac{1}{|V|}} = \tag{4}$$

$$\frac{P(X = x \mid i)}{\sum_{v \in V} P(X = x \mid v)} \propto P(X = x \mid i)$$

To find the concept $i$ that maximizes the posterior $P(V = i \mid X = x)$, we write:

$$\hat{i} = \arg\max_{i \in V} P(V = i \mid X = x) = \arg\max_{i \in V} P(X = x \mid i) = \arg\max_{i \in V} S_{\text{lemma}}(i, x).$$

Thus, under a uniform prior, maximizing the posterior probability is equivalent to maximizing the Lemma Similarity $\qquad \square$

## D.2  HYPERNYM SIMILARITY & COHYPONYM SIMILARITY

**Definition** (Hypernym Similarity). *Let $A(i) \subseteq V$ be the set of hypernyms of a concept $i \in V$. For a given image $x \in \mathcal{X}$, we define the* Hypernym Similarity *as:*

$$S_{hyper}(i, x) := P(X = x \mid A(i)) = \frac{1}{|A(i)|} \sum_{h \in A(i)} P(X = x \mid h).$$

**Definition** (Cohyponym Similarity). *Let $C(i) \subseteq V$ be the set of cohyponyms of a concept $i \in V$. For a given image $x \in \mathcal{X}$, we define the* Cohyponym Similarity *as:*

$$S_{cohyponym}(i, x) := P(X = x \mid C(i)) = \frac{1}{|C(i)|} \sum_{ch \in C(i)} P(X = x \mid ch).$$

*Hypernym Similarity* and *Cohyponym Similarity* represent the likelihood of observing $x$ under the average distribution of its ancestor and cohyponyms concepts respectively. Intuitively, they measure how well the image $x$ fits into the neighboring concepts either broader, more general semantic category represented by the ancestors of $i$ or similar, slightly different concepts from the same ancestor

of $i$. According further to Theorem 2, maximizing those similarities is proportional to minimizing distance between image space conditioned on a concept and image space conditioned on specific neighbor space, therefore better reflecting neighbors semantic properties and covering tree structure.

**Theorem 2.** *With large enough $S_{lemma}(i,x)$ to properly represent our concept,* $\max S_{hyper}(i,x)$ *and* $\max S_{cohyponym}(i,x) \propto \min_{P(X|A(i))} D_{KL}\big(P(X \mid i) \,\|\, P(X \mid A(i))\big)$.

*Proof.* The proof will be based for the ancestor case, however proving for cohyponyms is similar with only change of ancestor set to cohyponyms set.

Consider the KL divergence:

$$D_{\mathrm{KL}}\big(P(X \mid i) \,\|\, P(X \mid A(i))\big) = \sum_{x \in \mathcal{X}} P(X = x \mid i) \log \frac{P(X = x \mid i)}{P(X = x \mid A(i))}.$$

$$\arg\min_{P(X|A(i))} D_{\mathrm{KL}}\big(P(X \mid i) \,\|\, P(X \mid A(i))\big) \implies \frac{P(X = x \mid i)}{P(X = x \mid A(i))} \approx 1 \quad \forall x.$$

As $P(X = x \mid i)$ is fixed in precise setting, achieving $\frac{P(X=x|i)}{P(X=x|A(i))} \approx 1$ requires increasing $P(X = x \mid A(i))$ subject to large enough $P(X = x \mid i)$. Since $S_{\mathrm{hyper}}(i,x) = P(X = x \mid A(i))$, increasing $S_{\mathrm{hyper}}(i,x)$ for the most probable $x$ under $i$ reduces the KL divergence. $\square$

### D.3 SPECIFICITY

**Definition** (Specificity). *Let $C(i)$ be the set of cohyponyms of a concept $i \in V$. Define:*

$$P(X = x \mid C(i)) := \frac{1}{|C(i)|} \sum_{c \in C(i)} P(X = x \mid c).$$

*The* Specificity *of an image $x \in \mathcal{X}$ with respect to a concept $i \in V$ is:*

$$Spec(i,x) := \frac{P(X = x \mid i)}{P(X = x \mid C(i))}.$$

Specificity measures how much more likely it is that $x$ was generated by a concept $i$ compared to one of its cohyponyms. A high Specificity value indicates that the probability of $x$ under $i$ is significantly larger than under $C(i)$, the distribution over its cohyponyms. According to Theorems 3 and 4, Specificity highlights the uniqueness of the node representation by maximizing the distance to the cohyponyms nodes distribution and increasing the mutual information between the concept and image spaces. However, this metric relies on a high *Lemma Similarity* value; otherwise, the reflected uniqueness may be misleading due to poor alignment with the target node's distribution.

**Theorem 3.**
$$\max Spec(i,x) \propto \max D_{KL}(P(X \mid i)\|P(X \mid C(i)))$$

*Proof.*
$$\mathrm{Spec}(i,x) = \frac{P(X = x \mid i)}{P(X = x \mid C(i))}.$$

$$\log(\mathrm{Spec}(i,x)) = \log \frac{P(X = x \mid i)}{P(X = x \mid C(i))}.$$

$$D_{\mathrm{KL}}(P(X \mid i)\|P(X \mid C(i))) = \sum_{x} P(X = x \mid i) \log \frac{P(X = x \mid i)}{P(X = x \mid C(i))}.$$

For fixed $P(X = x \mid i)$, increasing $\frac{P(X=x|i)}{P(X=x|C(i))}$ for all $x$ (i.e., maximizing $\mathrm{Spec}(i,x)$) increases each $\log \frac{P(X=x|i)}{P(X=x|C(i))}$ term. Thus, the sum $D_{\mathrm{KL}}(P(X \mid i)\|P(X \mid C(i)))$ is maximized. $\square$

**Theorem 4.** *Let $V$ and $X$ be random variables over concepts and images.* $\max Spec(i,x) \forall v \in V$, $x \in X \propto \max I(V; X)$.

*Proof.*

$$I(V; X) = \sum_{v,x} P(V = v, X = x) \log \frac{P(X = x \mid V = v)}{P(X = x)}.$$

For $v = i$:

$$\frac{P(X = x \mid i)}{P(X = x)} = \frac{P(X = x \mid i)}{\sum_{v'} P(V = v')P(X = x \mid v')}.$$

Consider the uniform prior ($P(V = v') = \frac{1}{|V|}$):

$$P(X = x) = \frac{1}{|V|} \sum_{v'} P(X = x \mid v'), \quad P(X = x \mid C(i)) = \frac{1}{|C(i)|} \sum_{c \in C(i)} P(X = x \mid c).$$

$$\mathrm{Spec}(i, x) = \frac{P(X = x \mid i)}{P(X = x \mid C(i))}.$$

Increasing $\mathrm{Spec}(i, x) \propto \frac{P(X=x|i)}{P(X=x)}$, raising each term $P(V = i, X = x) \log \frac{P(X=x|i)}{P(X=x)}$, thus increasing $I(V; X)$. $\qquad\square$

## E  GPT-4 PROMPTS

We show the technical style prompt for GPT-4 in Figure 7 for more clarity on how images and user prompt were provided. We employed "gpt-4o-mini" version with API calls with images in high resolution.

The prompt for "gpt-4o-mini" to generate definitions for TaxoLLaMA3.1 predictions is presented below.

(1)

```
Write a definition for the word/phrase in one sentence.

Example:
Word: caddle
Definition: act as a caddie and carry clubs for a player

Word: bichon
Definition:
```

```
Please act as an impartial judge and evaluate the quality of the images provided by two AI
image assistants to the user prompt displayed below.

You should choose the assistant that provides an image that follows the user's instructions
better and reflects the user's prompt main concept better.
Your evaluation should consider factors such as image-text alignment, relevance, accuracy,
depth, and fidelity (overall image quality).
Begin your evaluation by comparing the two images and provide a short explanation.

Avoid any position biases and ensure that the order in which the responses were presented do
not influence your decision.

Do not allow the size of the images to influence your evaluation.
Do not favor certain names of the assistants.
Be as objective as possible.

After providing your explanation, output your final verdict by strictly following this format:
"[[A]]" if assistant A is better, "[[B]]" if assistant B is better, "[[C]]" for a tie and "
[[D]]" if both images are bad.

[User Prompt]

inherited disorder (a genetic condition passed down from parents to their offspring, often
resulting in physical or mental health problems)

[Start of the first image]

<img_a>

[End of the first image]

[Start of the first image]

<img_b>

[End of the first image]
```

Figure 7: Full prompt example for evaluating text-to-image assistants.

## F  TECHNICAL DETAILS

For text-to-image, we used the recommended generation parameters for each model, the Hugging-Face Diffusers library, and a single NVIDIA A100 GPU. All models were utilized in FP16 precision and produced images with resolutions of 512x512 or 1024x1024. Additionally, we experimented with prompting, adding definitions from the WordNet database to help with ambiguity resolution, as this has shown benefits for LLMs in the past Moskvoretskii et al. (2024b).

## G  ADDITIONAL FIGURES

In this appendix, we include graphs for our evaluation, which results outlined in the main text.

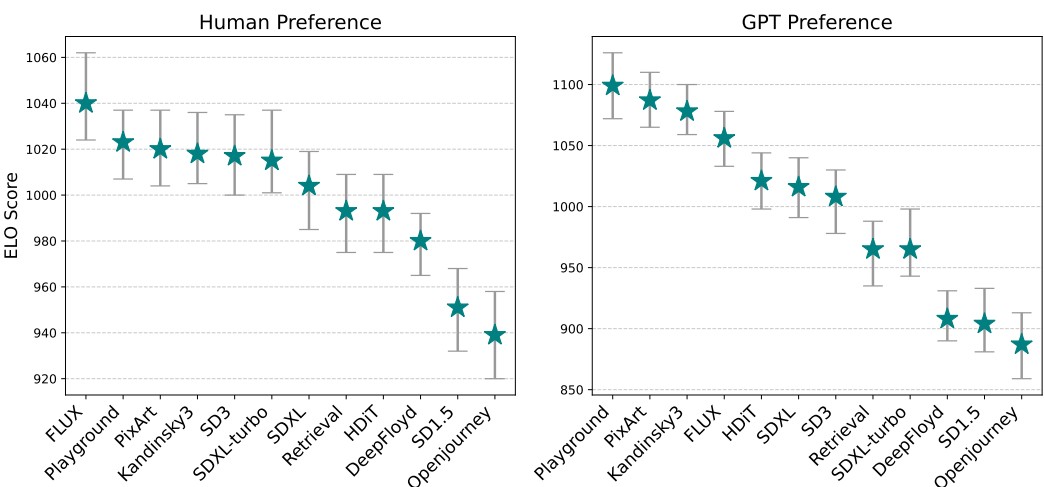

Figure 8: ELO scores for human and GPT4 preferences. Prompt did not include the definition.

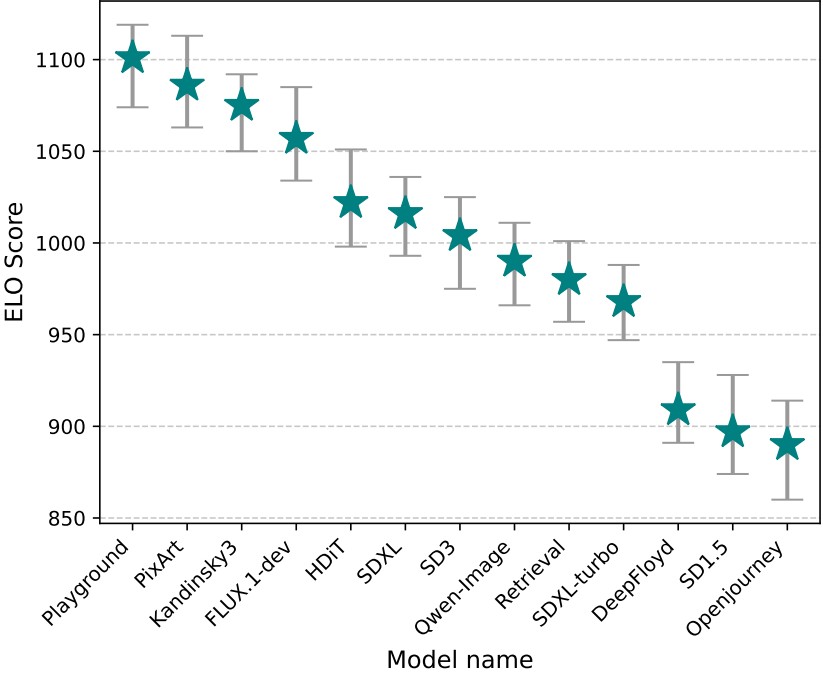

Figure 9: ELO scores for GPT4 preferences with Qwen-Image model. Prompt did not include the definition.

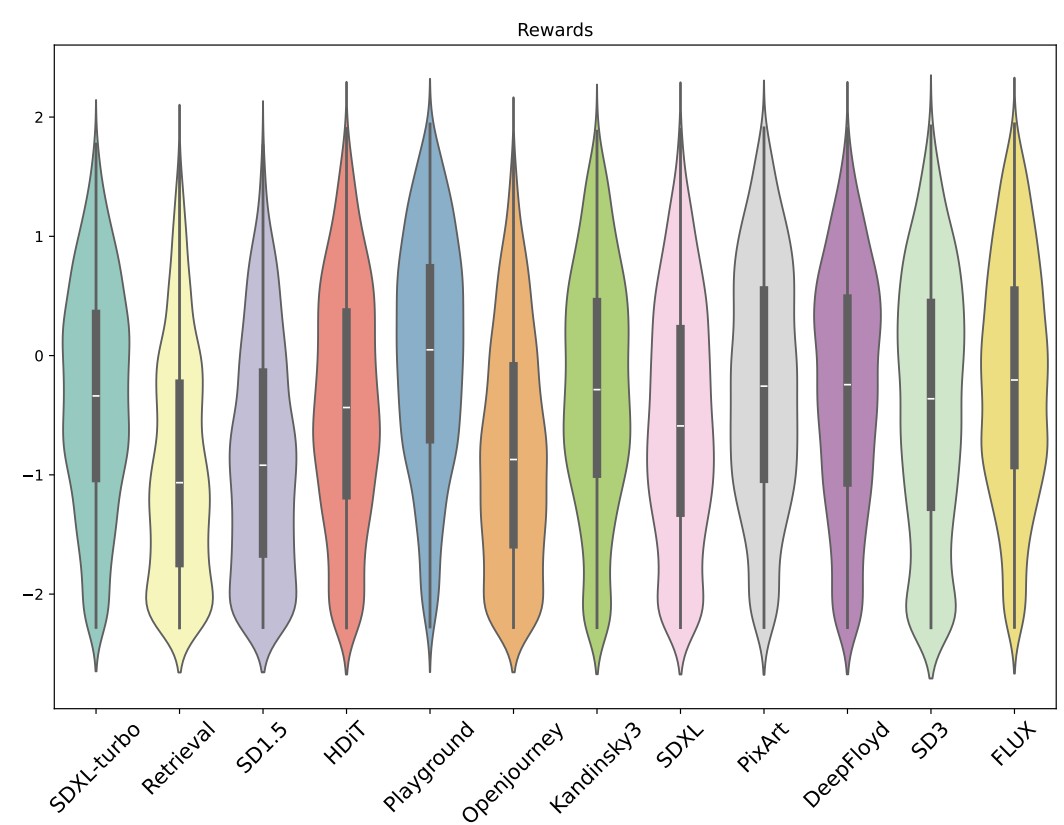

Figure 10: Distribution of rewards for each model, calculated with reward model described in Section 4

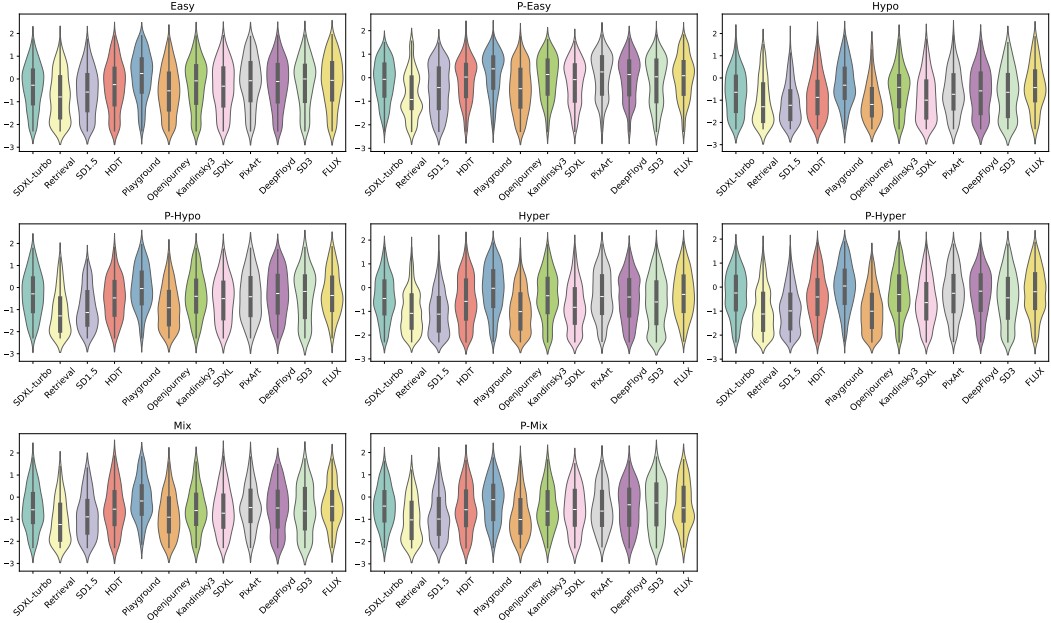

Figure 11: Distribution of rewards for each model across subsets, calculated with reward model described in Section 4

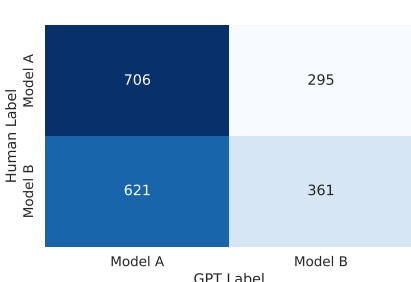

Figure 12: Confusion matrix for human and GPT preferences, excluding Tie labels to avoid distracting the analysis. GPT rarely assigns Ties, with fewer than 20 instances. The prompt included a definition.

| Model | Ground Truth | | | | Predicted | | | |
|---|---|---|---|---|---|---|---|---|
| | Easy | Hypo | Hyper | Mix | P-Easy | P-Hypo | P-Hyper | P-Mix |
| Playground | 1125 (+61/-59) | 1139 (+137/-111) | 1148 (+50/-56) | 1066 (+97/-105) | 1072 (+87/-85) | 1095 (+118/-89) | 1141 (+53/-59) | 1047 (+130/-105) |
| FLUX | 1013 (+65/-78) | 1104 (+153/-151) | 1088 (+48/-50) | 982 (+105/-125) | 1066 (+82/-60) | 967 (+131/-107) | 1025 (+46/-41) | 1096 (+144/-132) |
| PixArt | 1050 (+43/-67) | 1125 (+181/-100) | 1086 (+60/-40) | 1038 (+104/-80) | 1135 (+82/-66) | 1159 (+143/-95) | 1107 (+47/-49) | 1063 (+101/-136) |
| SDXL | 960 (+75/-72) | 1113 (+145/-149) | 1056 (+63/-61) | 1063 (+134/-128) | 1061 (+78/-67) | 1112 (+114/-86) | 1050 (+49/-48) | 1010 (+148/-120) |
| HDiT | 981 (+61/-61) | 955 (+100/-97) | 1004 (+44/-51) | 1053 (+122/-137) | 980 (+78/-59) | 1046 (+138/-86) | 1074 (+52/-61) | 965 (+148/-134) |
| Kandinsky3 | 1010 (+72/-70) | 1035 (+103/-101) | 998 (+51/-55) | 958 (+135/-82) | 1051 (+74/-55) | 999 (+100/-104) | 1043 (+48/-40) | 1005 (+102/-113) |
| Retrieval | 965 (+81/-76) | 884 (+98/-106) | 979 (+47/-58) | 1014 (+119/-105) | 953 (+65/-72) | 880 (+111/-135) | 995 (+51/-54) | 971 (+138/-137) |
| SD3 | 1056 (+78/-59) | 949 (+118/-99) | 962 (+41/-53) | 983 (+122/-113) | 997 (+49/-59) | 1090 (+123/-120) | 961 (+62/-57) | 1104 (+116/-85) |
| SDXL-turbo | 1004 (+86/-69) | 999 (+102/-93) | 957 (+49/-48) | 960 (+114/-148) | 917 (+74/-86) | 964 (+82/-117) | 969 (+46/-47) | 1025 (+110/-102) |
| DeepFloyd | 981 (+53/-63) | 909 (+76/-95) | 943 (+50/-43) | 1053 (+128/-110) | 931 (+56/-64) | 949 (+152/-158) | 941 (+40/-65) | 1036 (+105/-109) |
| Openjourney | 997 (+65/-59) | 849 (+102/-125) | 889 (+56/-62) | 962 (+97/-107) | 987 (+59/-77) | 880 (+87/-162) | 826 (+54/-55) | 825 (+125/-225) |
| SD1.5 | 852 (+69/-90) | 933 (+102/-120) | 885 (+58/-55) | 863 (+110/-115) | 842 (+65/-70) | 853 (+73/-130) | 864 (+48/-64) | 847 (+108/-133) |

Table 3: ELO score for GPT Preferences for subsets with definition in input.

| Model | Ground Truth | | | | Predicted | | | |
|---|---|---|---|---|---|---|---|---|
| | Easy | Hypo | Hyper | Mix | P-Easy | P-Hypo | P-Hyper | P-Mix |
| Playground | 1091 (+82/-47) | 1110 (+154/-123) | 1116 (+45/-42) | 1049 (+159/-101) | 1069 (+72/-66) | 1136 (+148/-118) | 1127 (+66/-53) | 1093 (+167/-104) |
| PixArt | 1037 (+77/-71) | 1137 (+110/-105) | 1113 (+49/-50) | 1094 (+103/-90) | 1122 (+86/-73) | 1048 (+122/-111) | 1081 (+45/-46) | 1076 (+123/-102) |
| Kandinsky3 | 1094 (+66/-74) | 1065 (+95/-110) | 1090 (+44/-57) | 1021 (+84/-107) | 1051 (+76/-67) | 1083 (+120/-72) | 1089 (+46/-52) | 1117 (+112/-120) |
| FLUX | 954 (+48/-65) | 1030 (+103/-113) | 1057 (+55/-57) | 1119 (+135/-106) | 1137 (+109/-64) | 1097 (+122/-100) | 1068 (+58/-46) | 1032 (+132/-128) |
| HDiT | 1028 (+58/-55) | 954 (+102/-129) | 1040 (+45/-41) | 1039 (+107/-102) | 1014 (+92/-56) | 1055 (+136/-86) | 1028 (+51/-48) | 835 (+118/-177) |
| SDXL | 1015 (+52/-61) | 939 (+104/-120) | 1029 (+58/-50) | 998 (+137/-156) | 1034 (+63/-64) | 1008 (+44/-45) | | 1033 (+133/-144) |
| SD3 | 1076 (+66/-76) | 951 (+108/-103) | 1006 (+50/-60) | 1099 (+129/-146) | 1014 (+62/-65) | 969 (+99/-100) | 964 (+34/-56) | 1078 (+124/-83) |
| Retrieval | 926 (+79/-70) | 996 (+99/-110) | 973 (+53/-43) | 1019 (+127/-109) | 917 (+81/-70) | 959 (+116/-104) | 994 (+49/-45) | 938 (+157/-138) |
| SDXL-turbo | 1045 (+76/-56) | 969 (+94/-94) | 906 (+56/-54) | 967 (+104/-110) | 950 (+66/-63) | 868 (+80/-146) | 980 (+43/-45) | 1073 (+115/-101) |
| DeepFloyd | 918 (+65/-63) | 900 (+131/-90) | 903 (+46/-60) | 888 (+108/-112) | 862 (+56/-86) | 976 (+99/-149) | 896 (+54/-61) | 980 (+125/-125) |
| SD1.5 | 870 (+71/-105) | 976 (+95/-134) | 888 (+58/-49) | 925 (+96/-115) | 894 (+69/-75) | 961 (+90/-80) | 905 (+34/-45) | 866 (+107/-187) |
| Openjourney | 940 (+76/-50) | 969 (+108/-135) | 874 (+43/-57) | 774 (+93/-139) | 930 (+60/-70) | 846 (+86/-129) | 854 (+47/-60) | 872 (+91/-154) |

Table 4: ELO score for GPT Preferences for subsets with no definition in input.

# H ADDITIONAL TABLES

In this section we provide the detailed tables for every metric evaluated in the paper.

**FID** results are demonstrated in Tables 8 and 10.

**IS** results are shown in Tables 9 and 8.

**Lemma Similarity** results are shown in Table 12

**Hypernym Similarity** results are shown in Table 11

**Cohyponym Similarity** results are shown in Table 15

**Specificity** results are shown in Table 14

**Reward model** p-values of Mann-Whitney test on comparing means shown in Table 16

**ELO Scores** for human and GPT labeling within each subset with and without definitions are shown in Tables 3, 4, 5 and 6.

| Model | Ground Truth | | | | Predicted | | | |
|---|---|---|---|---|---|---|---|---|
| | Easy | Hypo | Hyper | Mix | P-Easy | P-Hypo | P-Hyper | P-Mix |
| DeepFloyd | 1056 (+45/-38) | 957 (+72/-77) | 956 (+40/-27) | 945 (+89/-85) | 1013 (+60/-62) | 946 (+64/-90) | 993 (+28/-41) | 1027 (+55/-60) |
| Playground | 1055 (+37/-46) | 1102 (+96/-95) | 1078 (+36/-33) | 1094 (+81/-64) | 1017 (+67/-53) | 994 (+63/-62) | 1075 (+39/-33) | 1019 (+70/-68) |
| FLUX | 1050 (+48/-44) | 1105 (+128/-73) | 1088 (+37/-38) | 1048 (+104/-97) | 1218 (+77/-70) | 1070 (+74/-78) | 1051 (+29/-34) | 1044 (+90/-78) |
| SDXL-turbo | 1039 (+45/-46) | 987 (+57/-71) | 980 (+36/-41) | 945 (+89/-136) | 1018 (+53/-56) | 1027 (+69/-71) | 1033 (+36/-37) | 1053 (+79/-80) |
| SD3 | 1033 (+48/-41) | 1003 (+76/-71) | 1002 (+33/-35) | 996 (+58/-86) | 976 (+54/-47) | 976 (+68/-58) | 963 (+35/-32) | 959 (+57/-70) |
| SDXL | 1015 (+39/-36) | 1033 (+74/-113) | 1050 (+39/-31) | 1089 (+109/-85) | 980 (+54/-52) | 1060 (+73/-68) | 1035 (+32/-32) | 997 (+80/-71) |
| HDiT | 994 (+32/-43) | 946 (+70/-59) | 1031 (+33/-35) | 988 (+70/-76) | 1034 (+67/-70) | 1015 (+60/-58) | 1019 (+33/-38) | 953 (+80/-77) |
| PixArt | 990 (+43/-65) | 1027 (+69/-62) | 1082 (+33/-31) | 1050 (+78/-67) | 1030 (+59/-57) | 1052 (+71/-70) | 1036 (+36/-35) | 1075 (+72/-68) |
| Kandinsky3 | 961 (+42/-48) | 1059 (+81/-63) | 1014 (+38/-36) | 1005 (+87/-60) | 999 (+61/-87) | 992 (+51/-71) | 1063 (+40/-30) | 992 (+64/-97) |
| Retrieval | 960 (+50/-48) | 953 (+90/-83) | 940 (+39/-47) | 990 (+107/-104) | 890 (+62/-62) | 1023 (+111/-63) | 947 (+37/-45) | 1030 (+81/-90) |
| Openjourney | 941 (+45/-41) | 907 (+73/-66) | 885 (+37/-49) | 902 (+73/-102) | 928 (+79/-70) | 906 (+67/-70) | 874 (+42/-42) | 988 (+72/-73) |
| SD1.5 | 900 (+46/-59) | 914 (+72/-61) | 889 (+38/-36) | 942 (+76/-67) | 891 (+58/-70) | 933 (+68/-79) | 904 (+30/-34) | 857 (+61/-81) |

Table 5: ELO score for Human Preferences for subsets with definition in input.

| Model | Ground Truth | | | | Predicted | | | |
|---|---|---|---|---|---|---|---|---|
| | Easy | Hypo | Hyper | Mix | P-Easy | P-Hypo | P-Hyper | P-Mix |
| Playground | 1044 (+46/-33) | 1044 (+70/-68) | 1043 (+28/-30) | 1006 (+66/-67) | 1006 (+62/-55) | 972 (+76/-55) | 1039 (+32/-29) | 973 (+61/-69) |
| PixArt | 940 (+35/-41) | 1036 (+64/-74) | 1035 (+36/-30) | 1138 (+84/-84) | 1037 (+50/-46) | 954 (+68/-94) | 1020 (+26/-31) | 1041 (+74/-75) |
| Kandinsky3 | 979 (+47/-33) | 1060 (+97/-62) | 1027 (+39/-32) | 1034 (+63/-86) | 978 (+58/-49) | 1016 (+56/-72) | 1045 (+43/-37) | 998 (+73/-80) |
| SD3 | 1048 (+37/-54) | 967 (+67/-69) | 1026 (+32/-31) | 1074 (+80/-76) | 1044 (+69/-51) | 970 (+70/-52) | 996 (+27/-32) | 951 (+68/-76) |
| SDXL | 1022 (+46/-41) | 948 (+54/-52) | 1017 (+39/-27) | 939 (+114/-83) | 1016 (+44/-49) | 1026 (+92/-60) | 985 (+30/-24) | 1125 (+75/-75) |
| FLUX | 1011 (+51/-46) | 1015 (+75/-100) | 1008 (+42/-43) | 1012 (+92/-84) | 1144 (+75/-59) | 1102 (+79/-64) | 1043 (+32/-26) | 941 (+64/-83) |
| HDiT | 964 (+42/-41) | 988 (+61/-87) | 1001 (+28/-34) | 961 (+92/-93) | 1040 (+48/-41) | 949 (+69/-70) | 1001 (+36/-34) | 983 (+63/-88) |
| Retrieval | 1041 (+49/-49) | 1027 (+79/-72) | 995 (+32/-29) | 1051 (+71/-83) | 893 (+61/-57) | 1093 (+86/-73) | 976 (+37/-38) | 1027 (+96/-84) |
| SDXL-turbo | 992 (+48/-41) | 989 (+60/-65) | 984 (+37/-37) | 986 (+90/-88) | 1010 (+40/-56) | 1033 (+73/-68) | 1053 (+37/-23) | 1049 (+64/-66) |
| DeepFloyd | 1000 (+37/-40) | 969 (+62/-54) | 970 (+35/-39) | 1037 (+76/-78) | 959 (+40/-56) | 1013 (+64/-86) | 979 (+34/-31) | 963 (+57/-82) |
| SD1.5 | 968 (+52/-56) | 991 (+73/-69) | 958 (+36/-31) | 918 (+85/-79) | 935 (+48/-51) | 947 (+54/-65) | 948 (+25/-36) | 970 (+71/-63) |
| Openjourney | 983 (+48/-43) | 958 (+63/-52) | 930 (+32/-42) | 838 (+67/-96) | 931 (+63/-53) | 918 (+56/-63) | 908 (+33/-38) | 973 (+83/-95) |

Table 6: ELO score for Human Preferences for subsets with no definition in input.

| Model | Inception Score | FID |
|---|---|---|
| DeepFloyd | 19.6 | 62 |
| Kandinsky3 | 19.4 | 64 |
| PixArt | 19.8 | 73 |
| Playground | 20.9 | 71 |
| Openjourney | 15.4 | 68 |
| SD1.5 | 18.0 | 59 |
| SDXL-turbo | 10.9 | 89 |
| SD3 | 21.2 | 63 |
| HDiT | 18.2 | 67 |
| SDXL | 19.1 | 63 |
| FLUX | 20.9 | 68 |
| Retrieval | 19.1 | - |

Table 7: FID and IS metrics for different models on the full dataset without repetitions

| Model | IS | | FID | |
|---|---|---|---|---|
| | def | no_def | def | no_def |
| DeepFloyd | 19.6 | 12.1 | 62 | 131 |
| Kandinsky3 | 19.4 | 18.2 | 64 | 75 |
| PixArt | 19.8 | 18.8 | 73 | 73 |
| Playground | 20.9 | 20.0 | 71 | 74 |
| Openjourney | 15.4 | 13.9 | 68 | 73 |
| SD1.5 | 18.0 | 17.5 | 59 | 60 |
| SDXL-turbo | 10.9 | 12.1 | 89 | 65 |
| SD3 | 21.2 | 23.5 | 63 | 65 |
| HDiT | 18.2 | 20.2 | 67 | 85 |
| SDXL | 19.1 | 19.6 | 63 | 68 |
| FLUX | 20.9 | 22.0 | 68 | 72 |

Table 8: Comparing FID and IS for datasets with and without definition

| Model | Ground Truth | | | | Predicted | | | |
|---|---|---|---|---|---|---|---|---|
| | **Hypo** | **Hyper** | **Mix** | **Easy** | **Hypo** | **Hyper** | **Mix** | **Easy** |
| DeepFloyd | 8.2 | 9.3 | 6.6 | 12.9 | 8.3 | 11.3 | 6.8 | 7.3 |
| Kandinsky-3 | 7.4 | 10.3 | 6.6 | 12.8 | 7.8 | 11.7 | 7.0 | 8.4 |
| PixArt | 7.6 | 10.5 | 6.6 | 13.5 | 7.6 | 11.2 | 7.1 | 8.6 |
| Playground | 8.6 | 11.0 | 7.0 | 13.3 | 8.0 | 12.0 | 7.8 | 8.3 |
| Openjourney | 6.7 | 8.2 | 6.3 | 12.5 | 6.9 | 9.0 | 6.4 | 7.0 |
| SD1.5 | 7.1 | 9.0 | 6.9 | 12.9 | 7.4 | 9.5 | 6.6 | 8.1 |
| SDXL-turbo | 5.2 | 6.7 | 4.7 | 9.9 | 5.3 | 6.8 | 5.1 | 6.1 |
| SD3 | 8.4 | 9.5 | 7.2 | 13.3 | 8.1 | 12.0 | 7.3 | 8.8 |
| HDiT | 8.0 | 10.1 | 6.8 | 11.2 | 7.3 | 11.8 | 7.0 | 7.7 |
| SDXL | 7.6 | 10.4 | 7.1 | 12.8 | 7.5 | 11.4 | 7.3 | 8.1 |
| FLUX | 8.3 | 10.5 | 7.5 | 12.9 | 8.3 | 12.7 | 7.2 | 8.0 |
| Retrieval | 7.8 | 10.5 | 6.7 | 11.4 | 8.5 | 12.7 | 6.7 | 8.2 |

Table 9: Inception Score per subsets with definitions

| Model | Ground Truth | | | | Predicted | | | |
|---|---|---|---|---|---|---|---|---|
| | **Hypo** | **Hyper** | **Mix** | **Easy** | **Hypo** | **Hyper** | **Mix** | **Easy** |
| DeepFloyd | 195 | 134 | 191 | 128 | 220 | 219 | 147 | 193 |
| Kandinsky3 | 203 | 134 | 197 | 127 | 229 | 224 | 155 | 191 |
| PixArt | 203 | 143 | 191 | 134 | 229 | 230 | 163 | 203 |
| Playground | 198 | 143 | 188 | 130 | 234 | 225 | 163 | 196 |
| Openjourney | 200 | 145 | 197 | 127 | 225 | 228 | 159 | 198 |
| SD1.5 | 192 | 135 | 192 | 125 | 229 | 221 | 154 | 186 |
| SDXL-turbo | 207 | 158 | 211 | 146 | 235 | 226 | 170 | 203 |
| SD3 | 193 | 135 | 193 | 133 | 225 | 218 | 153 | 184 |
| HDiT | 201 | 141 | 187 | 126 | 226 | 225 | 160 | 198 |
| SDXL | 192 | 133 | 199 | 129 | 230 | 223 | 163 | 191 |
| FLUX | 163 | 132 | 186 | 139 | 165 | 115 | 182 | 171 |

Table 10: Fréchet Inception Distance Score per subsets with definitions

| Model | Ground Truth | | | | Predicted | | | |
|---|---|---|---|---|---|---|---|---|
| | **Easy** | **Hypo** | **Hyper** | **Mix** | **P-Easy** | **P-Hypo** | **P-Hyper** | **P-Mix** |
| HDiT | 0.65 | 0.60 | 0.61 | 0.61 | 0.57 | 0.59 | 0.59 | 0.60 |
| Retrieval | 0.63 | 0.56 | 0.57 | 0.60 | 0.54 | 0.57 | 0.56 | 0.60 |
| Kandinsky3 | 0.66 | 0.61 | 0.61 | 0.62 | 0.58 | 0.60 | 0.59 | 0.61 |
| Openjourney | 0.63 | 0.59 | 0.59 | 0.60 | 0.57 | 0.58 | 0.59 | 0.59 |
| DeepFloyd | 0.64 | 0.60 | 0.60 | 0.62 | 0.57 | 0.59 | 0.60 | 0.60 |
| SDXL-turbo | 0.67 | 0.62 | 0.62 | 0.63 | 0.59 | 0.61 | 0.60 | 0.61 |
| Playground | 0.66 | 0.60 | 0.61 | 0.61 | 0.57 | 0.59 | 0.59 | 0.60 |
| SDXL | 0.66 | 0.60 | 0.61 | 0.61 | 0.58 | 0.59 | 0.59 | 0.60 |
| PixArt | 0.65 | 0.60 | 0.60 | 0.62 | 0.57 | 0.59 | 0.59 | 0.61 |
| SD3 | 0.66 | 0.61 | 0.61 | 0.62 | 0.57 | 0.60 | 0.60 | 0.60 |
| SD1.5 | 0.64 | 0.60 | 0.60 | 0.61 | 0.57 | 0.58 | 0.60 | 0.59 |
| FLUX | 0.70 | 0.62 | 0.61 | 0.63 | 0.57 | 0.59 | 0.59 | 0.6 |

Table 11: Hypernym CLIPScore Across Different Subsets

| Model | Ground Truth | | | | Predicted | | | |
|---|---|---|---|---|---|---|---|---|
| | **Easy** | **Hypo** | **Hyper** | **Mix** | **Easy** | **Hypo** | **Hyper** | **Mix** |
| HDiT | 0.73 | 0.66 | 0.67 | 0.68 | 0.72 | 0.67 | 0.68 | 0.68 |
| Retrieval | 0.68 | 0.60 | 0.62 | 0.66 | 0.63 | 0.63 | 0.63 | 0.69 |
| Kandinsky3 | 0.73 | 0.67 | 0.67 | 0.69 | 0.73 | 0.68 | 0.68 | 0.69 |
| Openjourney | 0.71 | 0.66 | 0.66 | 0.68 | 0.71 | 0.67 | 0.66 | 0.69 |
| DeepFloyd | 0.71 | 0.67 | 0.66 | 0.69 | 0.70 | 0.67 | 0.66 | 0.69 |
| SDXL-turbo | 0.76 | 0.71 | 0.71 | 0.73 | 0.75 | 0.72 | 0.71 | 0.74 |
| Playground | 0.74 | 0.67 | 0.68 | 0.70 | 0.72 | 0.69 | 0.68 | 0.70 |
| SDXL | 0.73 | 0.67 | 0.67 | 0.69 | 0.73 | 0.69 | 0.68 | 0.70 |
| PixArt | 0.72 | 0.66 | 0.67 | 0.69 | 0.72 | 0.67 | 0.67 | 0.68 |
| SD3 | 0.73 | 0.68 | 0.67 | 0.70 | 0.72 | 0.69 | 0.68 | 0.70 |
| SD1.5 | 0.73 | 0.68 | 0.67 | 0.70 | 0.72 | 0.69 | 0.68 | 0.70 |
| FLUX | 0.71 | 0.65 | 0.66 | 0.68 | 0.72 | 0.68 | 0.67 | 0.69 |

Table 12: Lemma CLIPScore Across Different Subsets

| Model | CLIP-Score | Hypernym CLIP-Score | Cohyponym CLIP-Score | Specificity |
|---|---|---|---|---|
| HDiT | 0.69 | 0.60 | 0.58 | 1.2 |
| Retrieval | 0.64 | 0.57 | 0.56 | 1.16 |
| Kandinsky3 | 0.69 | 0.61 | 0.59 | 1.19 |
| Openjourney | 0.68 | 0.59 | 0.57 | 1.2 |
| DeepFloyd | 0.68 | 0.60 | 0.58 | 1.18 |
| SDXL-turbo | **0.72** | **0.62** | **0.60** | 1.23 |
| Playground | 0.70 | 0.60 | 0.58 | 1.22 |
| SDXL | 0.69 | 0.60 | 0.58 | 1.2 |
| PixArt | 0.68 | 0.60 | 0.58 | 1.19 |
| SD3 | 0.70 | 0.60 | 0.58 | 1.21 |
| SD1.5 | 0.69 | 0.59 | 0.57 | **1.23** |
| FLUX | 0.68 | 0.61 | 0.58 | 1.17 |

Table 13: Summary of CLIPscore Metrics Across Models

| Model | Ground Truth | | | | Predicted | | | |
|---|---|---|---|---|---|---|---|---|
| | **Easy** | **Hypo** | **Hyper** | **Mix** | **P-Easy** | **P-Hypo** | **P-Hyper** | **P-Mix** |
| HDiT | 1.21 | 1.14 | 1.15 | 1.16 | 1.34 | 1.18 | 1.18 | 1.18 |
| Retrieval | 1.17 | 1.11 | 1.13 | 1.14 | 1.24 | 1.15 | 1.15 | 1.18 |
| Kandinsky3 | 1.21 | 1.14 | 1.15 | 1.16 | 1.32 | 1.18 | 1.19 | 1.18 |
| Openjourney | 1.22 | 1.16 | 1.15 | 1.17 | 1.32 | 1.18 | 1.18 | 1.21 |
| DeepFloyd | 1.18 | 1.16 | 1.14 | 1.15 | 1.29 | 1.16 | 1.16 | 1.18 |
| SDXL-turbo | 1.23 | 1.18 | 1.18 | 1.20 | 1.34 | 1.22 | 1.22 | 1.24 |
| Playground | 1.24 | 1.16 | 1.17 | 1.20 | 1.34 | 1.21 | 1.20 | 1.21 |
| SDXL | 1.22 | 1.15 | 1.15 | 1.18 | 1.33 | 1.20 | 1.19 | 1.20 |
| PixArt | 1.20 | 1.14 | 1.15 | 1.16 | 1.34 | 1.17 | 1.18 | 1.17 |
| SD3 | 1.23 | 1.17 | 1.16 | 1.19 | 1.34 | 1.20 | 1.20 | 1.20 |
| SD1.5 | 1.24 | 1.19 | 1.18 | 1.20 | 1.34 | 1.22 | 1.21 | 1.23 |
| FLUX | 1.14 | 1.10 | 1.12 | 1.13 | 1.32 | 1.18 | 1.18 | 1.20 |

Table 14: Specificity Scores Across Different Models and Subsets

| Model | Ground Truth | | | | Predicted | | | |
|-------|------|------|-------|-----|--------|--------|---------|-------|
| | **Easy** | **Hypo** | **Hyper** | **Mix** | **P-Easy** | **P-Hypo** | **P-Hyper** | **P-Mix** |
| HDiT | 0.61 | 0.59 | 0.58 | 0.60 | 0.54 | 0.57 | 0.58 | 0.58 |
| Retrieval | 0.59 | 0.55 | 0.55 | 0.59 | 0.51 | 0.56 | 0.55 | 0.58 |
| Kandinsky3 | 0.61 | 0.59 | 0.59 | 0.61 | 0.55 | 0.58 | 0.59 | 0.59 |
| Openjourney | 0.59 | 0.58 | 0.57 | 0.59 | 0.54 | 0.57 | 0.57 | 0.57 |
| DeepFloyd | 0.61 | 0.58 | 0.58 | 0.61 | 0.55 | 0.57 | 0.58 | 0.59 |
| SDXL-turbo | 0.62 | 0.60 | 0.60 | 0.62 | 0.56 | 0.59 | 0.60 | 0.60 |
| Playground | 0.60 | 0.58 | 0.58 | 0.60 | 0.54 | 0.57 | 0.58 | 0.58 |
| SDXL | 0.61 | 0.58 | 0.58 | 0.60 | 0.55 | 0.58 | 0.58 | 0.60 |
| PixArt | 0.60 | 0.59 | 0.58 | 0.61 | 0.54 | 0.57 | 0.58 | 0.59 |
| SD3 | 0.61 | 0.59 | 0.58 | 0.60 | 0.54 | 0.58 | 0.58 | 0.59 |
| SD1.5 | 0.60 | 0.58 | 0.57 | 0.60 | 0.54 | 0.56 | 0.57 | 0.57 |
| FLUX | 0.63 | 0.59 | 0.59 | 0.61 | 0.54 | 0.57 | 0.57 | 0.58 |

Table 15: Cohyponym CLIPScore Across Different Subsets

| | SDXL-turbo | Retrieval | SD1.5 | HDiT | Playground | Openjourney | Kandinsky3 | SDXL | PixArt | DeepFloyd | SD3 | FLUX |
|---|---|---|---|---|---|---|---|---|---|---|---|---|
| SDXL-turbo | 0 | 0 | 0 | 0.003 | 0 | 0 | 0.036 | 0 | 0 | 0.027 | 0.029 | 0 |
| Retrieval | 0 | 0 | 0 | 0 | 0 | 0 | 0 | 0 | 0 | 0 | 0 | 0 |
| SD1.5 | 0 | 0 | 0 | 0 | 0 | 0.037 | 0 | 0 | 0 | 0 | 0 | 0 |
| HDiT | 0.003 | 0 | 0 | 0 | 0 | 0 | 0 | 0 | 0 | 0 | 0.702 | 0 |
| Playground | 0 | 0 | 0 | 0 | 0 | 0 | 0 | 0 | 0 | 0 | 0 | 0 |
| Openjourney | 0 | 0 | 0.037 | 0 | 0 | 0 | 0 | 0 | 0 | 0 | 0 | 0 |
| Kandinsky3 | 0.036 | 0 | 0 | 0 | 0 | 0 | 0 | 0 | 0.167 | 0.95 | 0 | 0 |
| SDXL | 0 | 0 | 0 | 0 | 0 | 0 | 0 | 0 | 0 | 0 | 0 | 0 |
| PixArt | 0 | 0 | 0 | 0 | 0 | 0 | 0.167 | 0 | 0 | 0.14 | 0 | 0 |
| DeepFloyd | 0.027 | 0 | 0 | 0 | 0 | 0 | 0.95 | 0 | 0.14 | 0 | 0 | 0 |
| SD3 | 0.029 | 0 | 0 | 0.702 | 0 | 0 | 0 | 0 | 0 | 0 | 0 | 0 |
| FLUX | 0 | 0 | 0 | 0 | 0 | 0 | 0 | 0 | 0 | 0 | 0 | 0 |

Table 16: P-value of Mann-Whitney mean differences test in rewards for models. Values below 0.000 are marked as 0. To identify the side of difference, refer to the violin plot in Figure 10

# I  MODELS' MISTAKE ANALYSIS

This analysis is made for generation without definitions.

All models struggle with depicting

a. abstract concepts;

b. nonfrequent and specific words ("orifice.n.01" with the lemma "rima");

c. notions of people with specific functional role ("holder.n.02" with the lemma "holder", for example).

Abstract concepts are handled with the following:

    1. Text in images (although not all models succeed in writing);

(a) Openjourney, prevention.n.01, prevention

(b) Openjourney, corporal_punishment.n.01, corporal punishment

(c) HDit, reform_movement.n.01, labor union

(d) SD3, murder.n.01, murder

(e) SD3, abstinence.n.02, abstinence

(f) PixArt, broadcast.n.01, headline

Figure 13: Text in images

    2. Abstract images;

Other unwanted behaviors for the purposes of illustrating taxonomies include

    1. Generating playing cards for the concepts (most seen in Openjourney, also present in SD1.5);

    2. Abstract ornamental circles (also most found in Openjourne, and some in SD1.5).

    3. Depicturing monsters when facing rare animal names (seen in Kandinsky3).

Most importantly, models struggle closer to the leaves of a taxonomy: they tend to create an image of a parent concept without necessary features of the child (see figure 21).

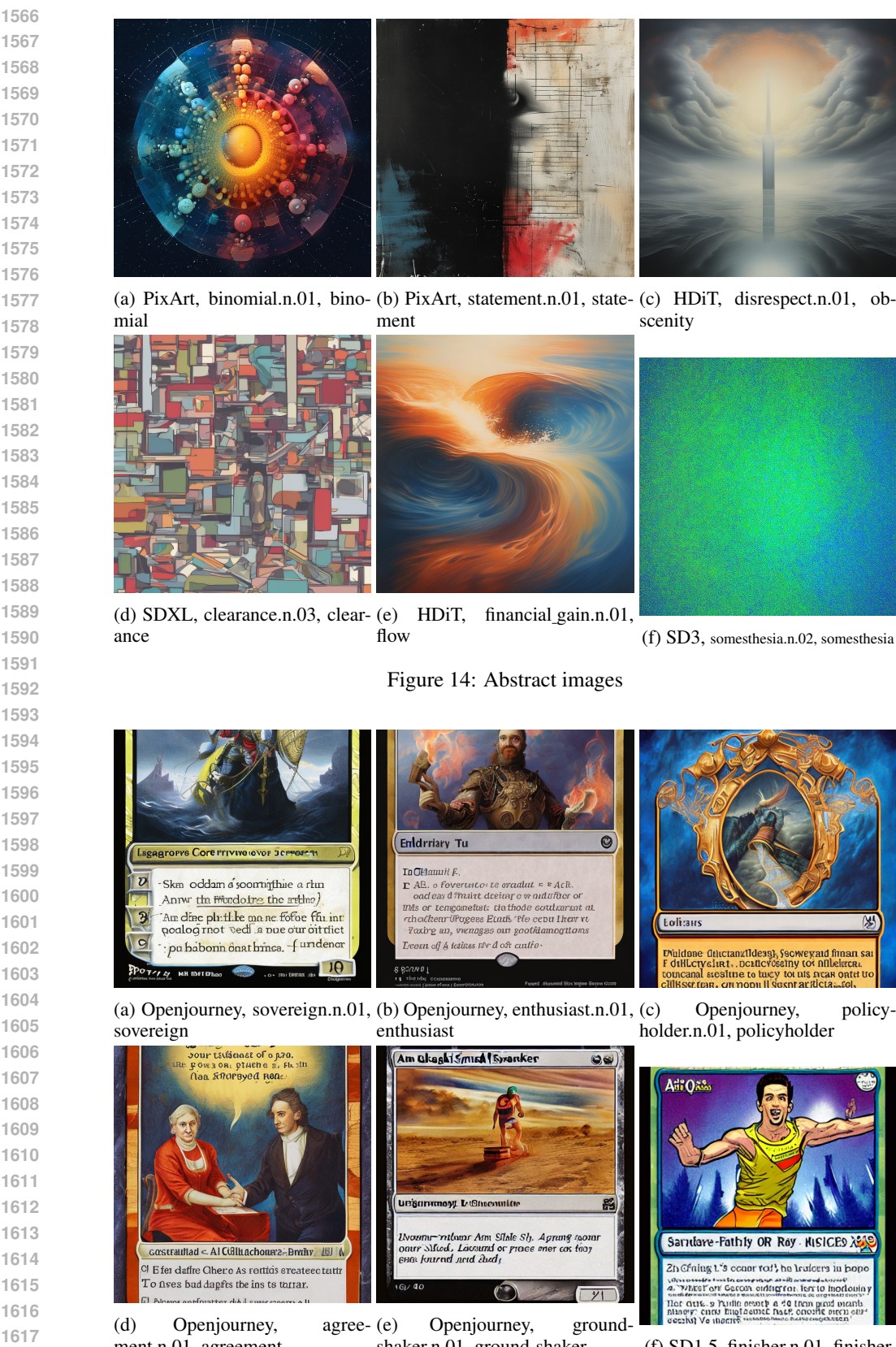

(a) PixArt, binomial.n.01, binomial

(b) PixArt, statement.n.01, statement

(c) HDiT, disrespect.n.01, obscenity

(d) SDXL, clearance.n.03, clearance

(e) HDiT, financial_gain.n.01, flow

(f) SD3, somesthesia.n.02, somesthesia

Figure 14: Abstract images

(a) Openjourney, sovereign.n.01, sovereign

(b) Openjourney, enthusiast.n.01, enthusiast

(c) Openjourney, policyholder.n.01, policyholder

(d) Openjourney, agreement.n.01, agreement

(e) Openjourney, ground-shaker.n.01, ground-shaker

(f) SD1.5, finisher.n.01, finisher

Figure 15: Playing cards in Openjourney and SD1.5

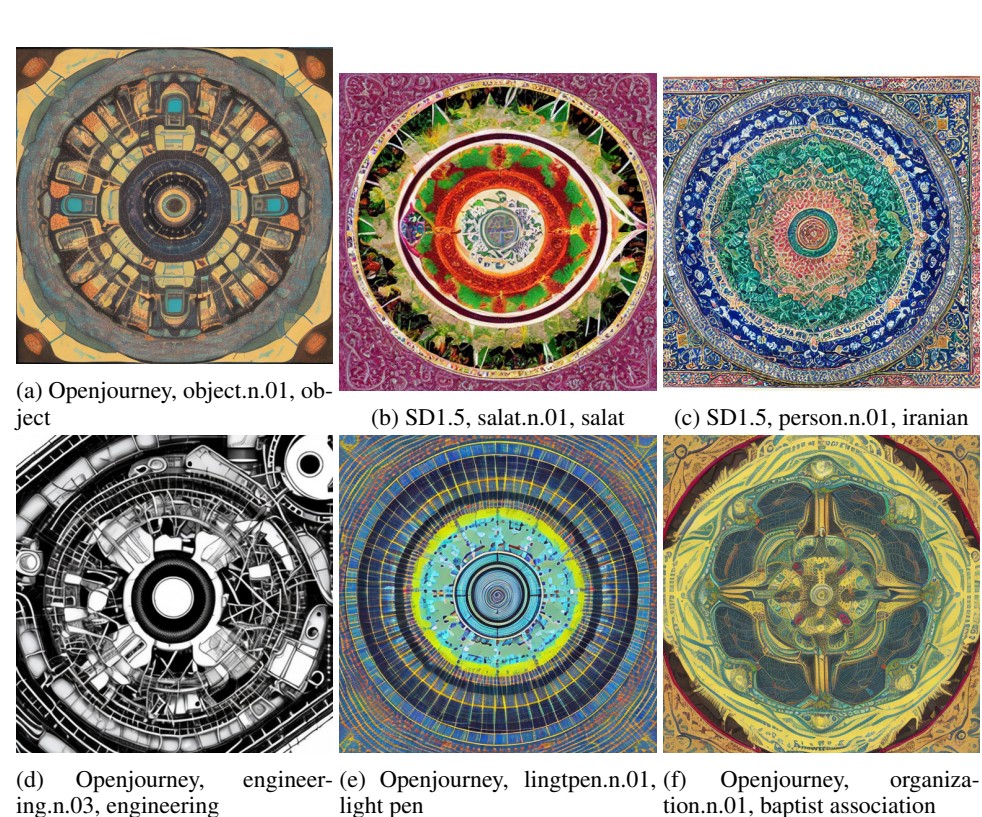

(a) Openjourney, object.n.01, object

(b) SD1.5, salat.n.01, salat

(c) SD1.5, person.n.01, iranian

(d) Openjourney, engineering.n.03, engineering

(e) Openjourney, lingtpen.n.01, light pen

(f) Openjourney, organization.n.01, baptist association

Figure 16: Abstract ornamental circles

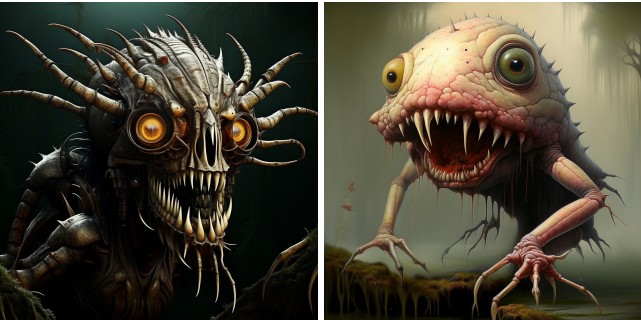

(a) Kandinsky3, acanthocephalan.n.01, acanthocephalan

(b) Kandinsky3, gelechiid.n.01, gelechiid

Figure 17: Monsters for rare animal names in Kandinsky

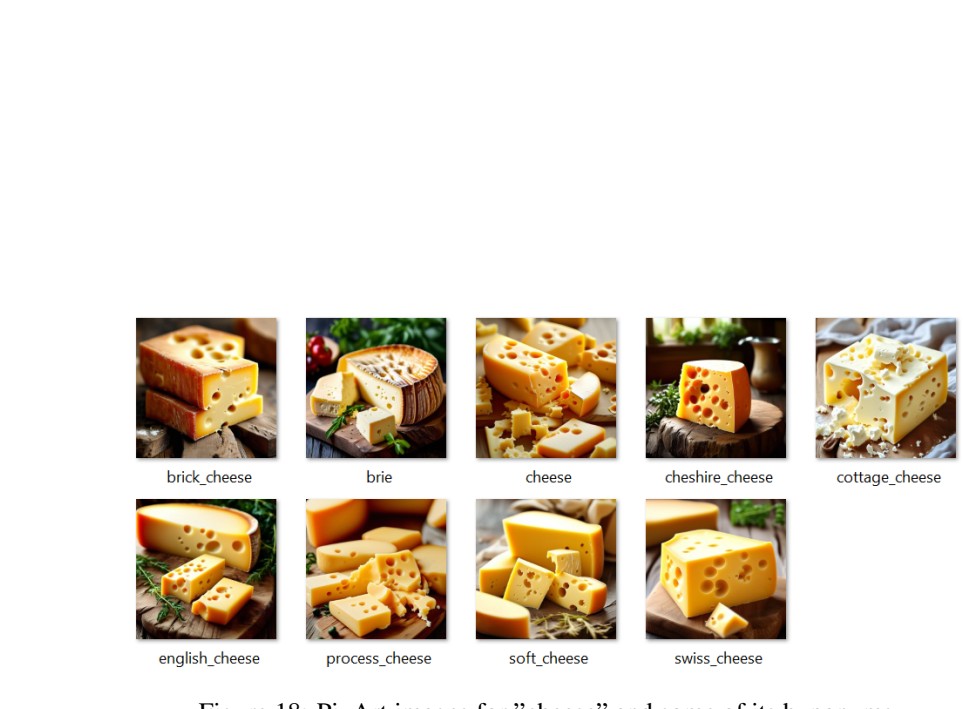

Figure 18: PixArt images for "cheese" and some of its hyponyms

## J DEMONSTRATION SYSTEM EXAMPLES

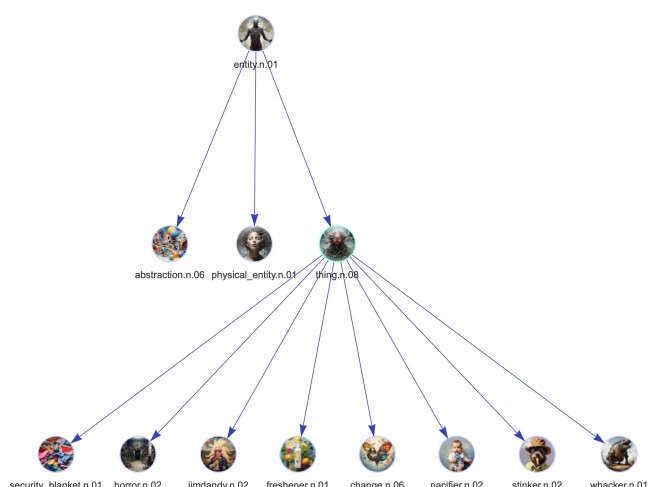

Figure 19: Subgraph starting from the root node "entity.n.01". Images are generated with the best TTI model.

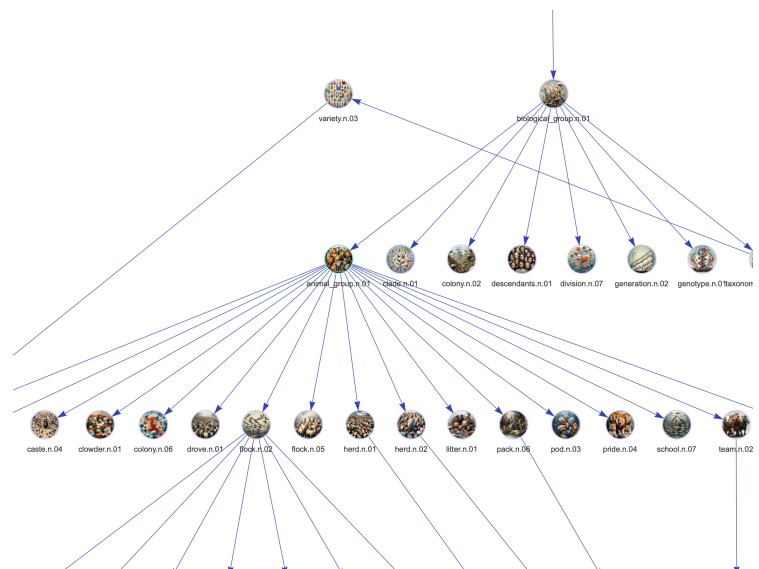

Figure 20: Subgraph starting from the node "biological_group.n.01". Images are generated with the best TTI model.

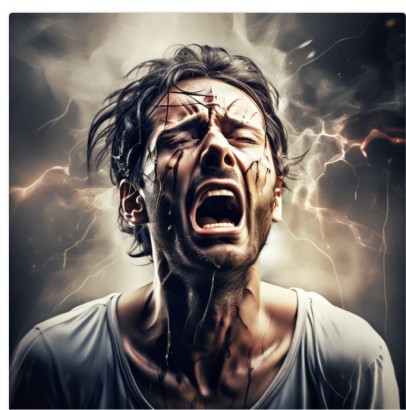

**biological_group.n.01**

biological group

a group of plants or animals

**pain.n.01**

pain,hurting

a symptom of some physical hurt or disorder

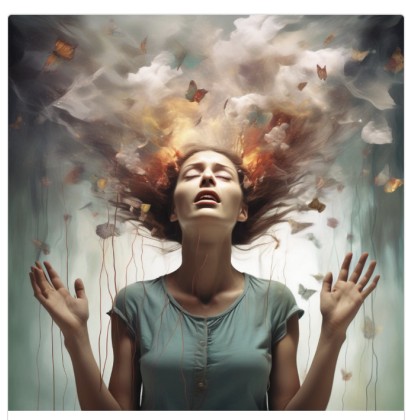

**feeling.n.01**

feeling

the experiencing of affective and emotional states

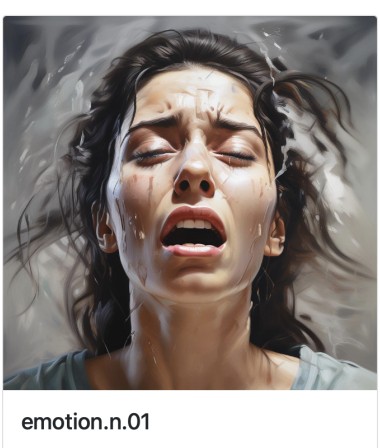

**emotion.n.01**

emotion

any strong feeling

Figure 21: Node descriptions from the demonstration system with the generated image using the best-performing model.

| Synset with definition | Lemma | Lemma with def |
|---|---|---|

**domestic_cat.n.01**
any domesticated member
of the genus Felis

**mouser.n.01**
a cat proficient at mousing

**angora.n.04**
a long-haired breed of cat
similar to the Persian cat

**egyptian_cat.n.01**
a domestic cat of Egypt

**alley_cat.n.01**
a homeless cat

**tiger_cat.n.01**
a cat having a striped coat

**tomcat.n.01**
a male cat

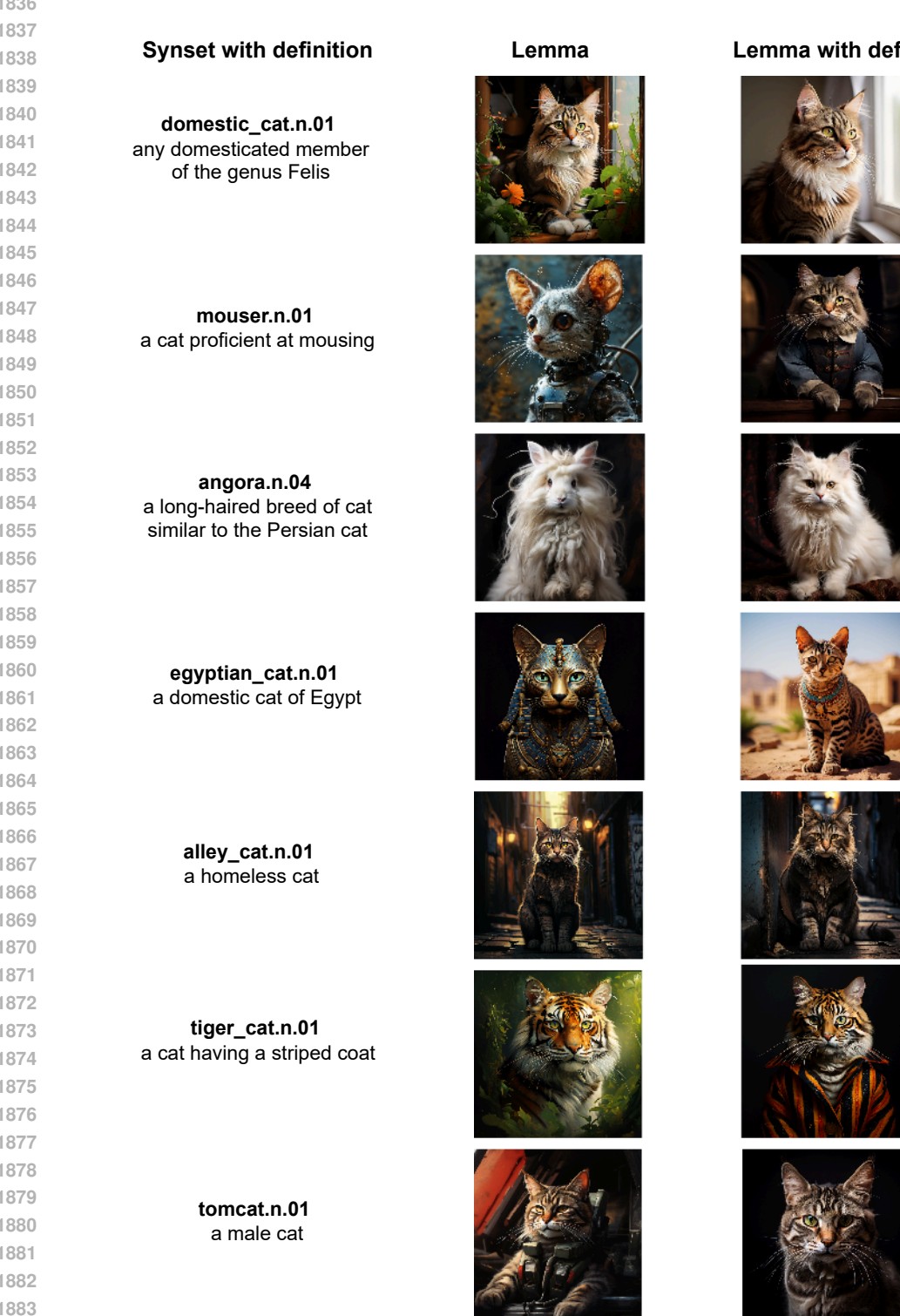

Figure 22: Examples of the generations for the 'domestic_cat.n.01' hyponyms using playground-v2.5-1024px-aesthetic model. The examples illustrate common confusion patterns without introducing an additional labeling step: mouser has features from "Mouser Electronics" and a mouse; Angora is mixed within a rabbit and has wool patterns; tiger cat either looks like a tiger or has a striped coat; tomcat is confused with aircraft and has made a cat a military pilot.

