# OpenReview forum: "Do I look like a "cat.n.01" to you? A Taxonomy Image Generation Benchmark"
_ICLR.cc/2026/Conference — Submitted to ICLR 2026_

### Official Review · Reviewer_qiUN · 2025-10-29

**Soundness:** 3
**Presentation:** 3
**Contribution:** 3
**Rating:** 4
**Confidence:** 5

**Summary:**

This paper introduces a comprehensive benchmark for evaluating text-to-image (T2I) models on taxonomy-based concept generation. The benchmark, called Taxonomy Image Generation, measures how well models can visualize WordNet concepts of varying abstraction levels.
It includes: three datasets (Easy Concepts, Random WordNet split, and LLM-generated predictions), 12 models (e.g., SDXL, Playground, FLUX, PixArt, etc.), nine metrics including ELO-based pairwise evaluation, reward models, and new taxonomy-driven similarity measures derived from KL divergence and mutual information.
The authors use both human and GPT-4-based pairwise evaluations, finding strong correlation (Spearman ≈ 0.9) between them. Results show Playground and FLUX consistently outperform others. The study concludes that T2I models can effectively visualize hierarchical lexical concepts, extending the visual dimension of WordNet beyond ImageNet.

**Strengths:**

1.Novel task definition linking taxonomy and image generation.

2.Methodological depth — integrates human, GPT-based, and reward-model evaluations.

**Weaknesses:**

1.The evaluation heavily depends on GPT-4 judgments, which could introduce unquantified biases.

2.Missing ablation studies (e.g., comparing against pure CLIP-based baselines).

3.Do the authors plan to extend this benchmark to multilingual or cross-lingual taxonomies?

4.How is mutual-information-based similarity normalized across models?

**Questions:**

see weaknesses

---

> ### Author Response · Authors · 2025-11-19
> **Answer to Reviewer qiUN**
>
> ### **W1 The evaluation heavily depends on GPT-4 judgments, which could introduce unquantified biases.**
>
> We thank the reviewer for raising this concern, but the evaluation does not heavily depend on GPT-4 judgments. GPT-4 is only one of the nine metrics we report, and it is used as a single comparative signal, not as the core evaluation mechanism. Our analysis integrates a diverse set of metrics, including human annotations, a reward model, and multiple CLIP-based similarity measures, each capturing different aspects of the semantic alignment between concepts and images. The conclusions in the paper do not rely on GPT-4 alone; rather, we consistently observe similar model rankings and trends across these heterogeneous metrics. We clarified this point in the revision to avoid the impression that the evaluation is dominated by GPT-4 when listing contributions in Introduction and 4.1.
>
> ### **W2 Missing ablation studies (e.g., comparing against pure CLIP-based baselines).**
>
> We thank the reviewer for the suggestion, however, we believe it is a misunderstanding.  We already include pure CLIP-based baselines in our evaluation: it is called Lemma Similarity and it reflects how well the image aligns with the lemma’s textual description, without the contribution of taxonomic relations. If the Reviewer refers to the CLIP + Image generator models, we agree that these would be interesting points of comparison. However, given the breadth of existing CLIP-generator combinations, it is not feasible to evaluate them exhaustively within the scope of this work.
>
> We would be grateful if the Reviewer could indicate which particular CLIP-based image-generation method they consider most relevant. We would be happy to incorporate an additional experiment with the suggested model in our revised version, provided computational constraints permit it.
>
> ### **Q3.Do the authors plan to extend this benchmark to multilingual or cross-lingual taxonomies?**
>
> We appreciate the reviewer’s question regarding multilingual or cross-lingual extensions. While this is an important future direction, extending the benchmark beyond WordNet requires substantial additional work.
> - **ConceptNet**, although multilingual, is not suitable for our setting: it **does not include images**, and its nodes are noisy because they integrate terms from heterogeneous sources (including Wikidata) without human curation or sense-level disambiguation. This makes it incompatible with the fine-grained linguistic and taxonomic analysis performed in our benchmark.
> - **Wikidata** provides images, however, its hierarchical relations (“subclass of,” “instance of”) are not linguistically curated. The resulting taxonomy is highly heterogeneous in granularity and does not support the controlled taxonomic evaluation we target.
>
> Extending the benchmark to Wikidata, ConceptNet, or other multilingual resources would require building new alignment procedures, filtering noisy entries, and defining new visual grounding standards essentially a **separate research effort**. We added it as future work in Limitations in Appendix A.
>
> ### **Q4. How is mutual-information-based similarity normalized across models?**
>
> We thank the reviewer for the question. Our mutual-information–based similarity is not model-specific and does not require per-model normalization. The MI is computed on the joint distribution of probability scores over images, where each model produces a normalized probability vector (via softmax) for the same set of candidate images. The MI is therefore naturally on a comparable scale across models because the inputs are probability distributions with the same support and normalization.
>
> ### **All in all:**
>
> We thank the reviewer again for the valuable feedback that helps us improve the paper. Therefore, in case the reviewer thinks that our paper deserves a better chance for publication, we would be extremely grateful for improving their score, taking into consideration the answers above.

---

> > ### Author Response · Authors · 2025-11-25
> > **Invitation to discussion**
> >
> > Dear Reviewer,
> >
> > As the rebuttal deadline is quickly approaching, this is a gentle reminder to please review and respond to the authors rebuttal comments.
> >
> > If you have any concerns, clarifications, or updates to your assessment after reading the response, please share them as soon as possible so we can address them promptly before the deadline.
> >
> > If you feel that our responses adequately address the concerns raised in your initial review, we would be extremely grateful for improving the score.
> >
> > Thank you very much for your time and contributions.
> >
> > Authors.

---

> > > ### Comment · Reviewer_qiUN · 2025-11-27
> > > **Official response of Reviewer qiUN**
> > >
> > > The reviewer likes the response and thus adjusts the score accordingly.

---

### Official Review · Reviewer_zHqq · 2025-10-31

**Soundness:** 2
**Presentation:** 2
**Contribution:** 2
**Rating:** 4
**Confidence:** 3

**Summary:**

The paper proposes a “taxonomy-aware” benchmark for text-to-image (T2I) generation: given WordNet synsets (e.g., cat.n.01), models must produce an image that faithfully depicts the target concept in its taxonomic sense. The authors build several test splits (common concepts, random WordNet nodes, and LLM-expanded concepts), evaluate multiple zero-shot T2I systems with prompts with/without textual definitions, and score results via pairwise preference (ELO/BT), reward models, taxonomy-aware CLIP similarities (lemma/hypernym/co-hyponym, specificity), plus FID/IS. Key findings: model rankings differ from mainstream T2I leaderboards; generation generally beats retrieval; adding a textual definition improves preference alignment.

**Strengths:**

- Well-motivated task: focuses on whether models grasp taxonomic meaning rather than just surface text-image alignment.

- Diverse evaluation protocol that mixes human/GPT-based preference with taxonomy-aware automatic metrics.

- Clear experimental settings (with/without definition) that isolate the impact of definitional context.

- Useful analysis showing generation > retrieval under this task design.

- Potentially valuable artifact: standardized synset prompts/splits that others can reuse.

**Weaknesses:**

- Missing 2025-era baselines: the model panel omits strong recent systems (e.g., Qwen-Image family, the latest GPT multimodal generators, and other 2025-vintage open models). This limits the external validity of the conclusions.

- Single-image, point-accuracy perspective is narrow: current metrics say “right vs. wrong” at the target synset but don’t quantify how wrong. A hierarchy-distance sensitive measure (path-length / information-content similarity such as Wu-Palmer, Jiang–Conrath, Lin, or a taxonomic cross-entropy) would reveal whether errors drift to ancestors vs. co-hyponyms.

- Insufficient error visualization: no compact one-page confusion view for co-hyponyms/siblings, making it hard to see systematic fine-grained failures (e.g., sibling breeds within a genus).

- (Secondary) Retrieval baseline appears underpowered; fairness controls (equal-time/equal-compute) and rank uncertainty (CIs, Kendall τ stability) are under-reported.

**Questions:**

- Model coverage: Can you add at least one 2025 frontier model and one 2025 open-source model (e.g., Qwen-Image-X, latest GPT-image, etc.) and re-report the main tables? If closed models are hard to access, provide an equal-time and equal-compute subset to ensure fair comparison.

- Hierarchy-distance metric: Please report a taxonomic distance–aware score (e.g., average shortest-path or an IC-based similarity from WordNet) and break down errors into ancestor vs. sibling vs. distant categories. A simple plot of depth vs. accuracy and sibling density vs. confusion rate would be very informative.

- One-page error analysis: Add a co-hyponym confusion matrix per super-class (e.g., within feline), with 4–6 representative visual examples annotated by the discriminative attributes that caused confusion (pattern, morphology, context).

---

> ### Author Response · Authors · 2025-11-20
> **Answer to Reviewer zHqq (part 1)**
>
> ### **W1 on 2025 frontier models**
>
> We appreciate the Reviewer’s point regarding the rapid emergence of 2025-era models. The pace of progress in multimodal generation is extremely fast, and it could be even impossible maintain exhaustively up-to-date coverage and the computational cost associated with evaluating each new release. However, we have started the experiments with Qwen-Image  (and will update you once they are finished) and will provide the benchmarking dataset for reproducibility: https://anonymous.4open.science/r/TIGB-E002/data/main_bench.csv
>
> Nevertheless, we would like to emphasize that the value of the benchmark does not depend on incorporating every latest model. The purpose of our benchmark is not to track the absolute state of the art in multimodal generation at every release cycle, but to **provide a stable, conceptually grounded evaluation framework** using a wide range of evaluation metrics. Therefore, adding an additional experiment would not influence the contributions of our paper.
>
> ### **W2 on Hierarchy-distance metric**
> We thank the reviewer for this suggestion. In our current setup, models **output images rather than explicit WordNet synset labels**, so a taxonomic-distance metric is not directly available. To compute shortest-path or IC-based distances in the WordNet hierarchy, one would first need a predicted synset for each generated / retrieved image. This would require adding an additional vision classifier (or a separate human labeling step), which may introduce extra source of noise and complexity in the main experiments. However, we see this as a substantial additional layer on top of our present evaluation and therefore beyond the scope of this paper as future work.
>
> ### **W3 on Insufficient error visualization**
>
> Error analysis is presented in Appendix I. Upon request, we have provided an example of such generations for the feline siblings (Figure 21, Appendix I, last page), however, this example is primarily demonstrative: since our evaluation does not involve automatic classification or human synset labeling of the generated images (as discussed above), we cannot construct a fully quantitative confusion matrix in the standard sense. The examples  illustrate common confusion patterns without introducing an additional labeling step: mouser has features from ``Mouser Electronics'' and a mouse; Angora is mixed within a rabbit and has wool patterns; tiger cat either looks like a tiger or has a striped coat; tomcat is confused with aircraft and has made a cat a military pilot. We also make this clarification explicit in the revision.
>
> ### **W4**
>
> We would like to clarify that the retrieval baseline is fairly configured and matches the intended comparison strength. As described in Appendix B.3, we rely on Wikimedia Commons as the image source because it provides the largest openly licensed collection that permits redistribution [2]. Using other large-scale image sources (e.g., Google Images, commercial APIs) would introduce copyright and licensing barriers, making them unsuitable for real-world use or even for training new models.
>
> We use Wikimedia’s built-in search engine, which is the only retrieval interface available in this setting, and we consistently take the top-1 returned image. The retrieval query uses the same prompt template as generation models, ensuring a fair and controlled comparison.
>
> We also report confidence intervals where applicable. For example, Figure 4 shows that retrieval ranks significantly lower than generation models even when confidence intervals are taken into account.
>
> The observed underperformance is due to inherent limitations of retrieval, not a weakness or unfairness of our setup:
> - Retrieval systematically struggles with rare or previously unseen concepts [1,3].
> - Retrieval is bounded by database contents and cannot produce images for missing or novel nodes.
> - Retrieval from large external image databases is often legally constrained: ImageNet is restricted to non-commercial use, while LAION and Google Search results are subject to copyright and personality-rights limitations.

---

> > ### Author Response · Authors · 2025-11-20
> > **Answer to Reviewer zHqq**
> >
> > ### **Q1: Same as W1**
> >
> > Sure, it is possible and we have started the experiments with Qwen-Image  (and will update you once they are finished) and provide the benchmarking dataset for reproducibility: https://anonymous.4open.science/r/TIGB-E002/data/main_bench.csv
> >
> > Nevertheless, we would like to emphasize that the value of the benchmark does not depend on incorporating every latest model. The purpose of our benchmark is not to track the absolute state of the art in multimodal generation at every release cycle, but to **provide a stable, conceptually grounded evaluation framework** using a wide range of evaluation metrics. Therefore, adding an additional experiment would not influence the contributions of our paper.
> >
> > ### **Q2: Same as W2**
> >
> > We agree that a hierarchy-aware analysis (e.g., ancestor vs. sibling vs. distant confusions, depth vs. performance) would be informative, and we have added this as an explicit avenue for future work. A natural extension would be to assign WordNet synsets to each image (via either human annotation or a fixed, off-the-shelf classifier) and then compute average shortest-path / IC-based distances, as well as bucket errors into ancestor / sibling / distant categories to compare models. However, we see this as a substantial additional layer on top of our present evaluation and therefore beyond the scope of this paper.
> >
> > ### **Q3: Same as W3**
> >
> > We have provided an example of such generations for the feline siblings, however, this analysis is primarily demonstrative: since our evaluation does not involve automatic classification or human synset labeling of the generated images (as discussed above), we cannot construct a fully quantitative confusion matrix in the standard sense. The examples  illustrate common confusion patterns without introducing an additional labeling step: mouser has features from ``Mouser Electronics'' and a mouse; Angora is mixed within a rabbit and has wool patterns; tiger cat either looks like a tiger or has a striped coat; tomcat is confused with aircraft and has made a cat a military pilot. We also make this clarification explicit in the revision.
> >
> >
> > ### **Bibliography**
> >
> > [1] Jones, Shawn M., and Diane Oyen. "Abstract images have different levels of retrievability per reverse image search engine." European Conference on Computer Vision. Cham: Springer Nature Switzerland, 2022.
> >
> > [2] Ferrada, Sebastián, et al. "Querying Wikimedia images using Wikidata facts." Companion Proceedings of the The Web Conference 2018. 2018.
> >
> > [3] Parashar, Shubham, et al. "The neglected tails in vision-language models." Proceedings of the IEEE/CVF Conference on Computer Vision and Pattern Recognition. 2024.

---

> > > ### Author Response · Authors · 2025-11-25
> > > **Invitation for discussion**
> > >
> > > Dear Reviewer,
> > >
> > > We have added Figure 9 with the GPT ELO scores including Qwen-Image. Here is the table with the ELO scores below:
> > >
> > > | | | |
> > > |-|-|-|
> > > playgroundai_playground-v2.5-1024px-aesthetic      | score: 1101  | 95% CI:  (-18, 27)  |
> > > PixArt-alpha_PixArt-Sigma-XL-2-512-MS              | score: 1086  | 95% CI:  (-27, 23)  |
> > > kandinsky-community_kandinsky-3                    | score: 1075  | 95% CI:  (-17, 25)  |
> > > black-forest-labs_FLUX.1-dev                       | score: 1057  | 95% CI:  (-28, 23)
> > > Tencent-Hunyuan_HunyuanDiT-v1.2-Diffusers          | score: 1022  | 95% CI:  (-29, 24)
> > > stabilityai_stable-diffusion-xl-base-1.0           | score: 1016  | 95% CI:  (-20, 23)
> > > stabilityai_stable-diffusion-3-medium-diffusers    | score: 1004  | 95% CI:  (-21, 29)
> > > Qwen_Qwen-Image                                    | score:  990  | 95% CI:  (-21, 24)
> > > retrieval                                          | score:  980  | 95% CI:  (-21, 23)
> > > stabilityai_sdxl-turbo                             | score:  968  | 95% CI:  (-20, 21)
> > > DeepFloyd_IF-I-XL-v1.0                             | score:  909  | 95% CI:  (-26, 18)
> > > runwayml_stable-diffusion-v1-5                     | score:  897  | 95% CI:  (-31, 23)
> > > prompthero_openjourney                             | score:  890  | 95% CI:  (-24, 30)
> > >
> > > As the rebuttal deadline is quickly approaching, this is a gentle reminder to please review and respond to the authors rebuttal comments.
> > >
> > > If you have any concerns, clarifications, or updates to your assessment after reading the response, please share them as soon as possible so we can address them promptly before the deadline.
> > >
> > > If you feel that our responses adequately address the concerns raised in your initial review, we would be extremely grateful for improving the score.
> > >
> > > Thank you very much for your time and contributions.
> > >
> > > Authors.

---

### Official Review · Reviewer_KcuZ · 2025-10-31

**Soundness:** 2
**Presentation:** 1
**Contribution:** 2
**Rating:** 2
**Confidence:** 3

**Summary:**

The paper proposes "Taxonomy Image Generation" as a novel benchmark task, aiming to evaluate text-to-image (T2I) models on their ability to visualize concise taxonomic concepts from WordNet. Unlike standard T2I tasks that use detailed descriptive prompts, this benchmark focuses on short, often abstract lemmas (e.g., "landscape"). The study evaluates 12 models and one retrieval baseline using 9 metrics, including novel taxonomy-specific scores grounded in KL Divergence (Hypernym/Cohyponym Similarity, Specificity).

**Strengths:**

- The paper identifies a gap in current T2I evaluation: standard benchmarks rely on rich, detailed descriptions (like DiffusionDB ), which mask model weaknesses in handling the concise, abstract prompts typical of taxonomic entries (Figure 1 ).

- The attempt to derive taxonomy-specific metrics (Hypernym Similarity, Specificity) grounded in the hierarchical structure of WordNet is a conceptually interesting move beyond standard FID/IS
.

**Weaknesses:**

- While the authors note that taxonomic prompting is difficult due to brevity (Figure 1 ), their benchmark heavily relies on adding explicit definitions to prompts to achieve good performance. This shifts the task from evaluating whether a model "knows" a taxonomic concept (e.g., "cigar_lighter.n.01") to evaluating standard instruction following. The significant drop in Human-GPT correlation (0.92 to 0.73) when definitions are removed indicates the definition, not the concept, is carrying the evaluation.

- The human evaluation section is quite sparse. While it mentions using 4 experts in computational linguistics for 3370 image pairs, it lacks critical reproducibility details. It does not specify the recruitment platform, compensation, or crucially, how many raters evaluated each pair. A total of 4 raters for a major benchmark is also arguably too small a pool to ensure diversity of perspective. Details about the task are also missing.

- The proposed 'Specificity' metric appears empirically flawed. Models that perform poorly in human preference (e.g., SD1.5) rank highly in Specificity. A metric that inversely correlates with human judgment and requires "interpretation" to be useful is not a strong contribution.

- Several presentation and formatting Issues:
  - The submission does not follow standard ICLR formatting (e.g., incorrect bottom margins)
  - Citations are frequently malformed and integrated poorly into the text flow (e.g., missing parentheses or awkward placement)
  - The results section is extremely short and most of the results were left in the Appendix

**Questions:**

- How do you validate that your CLIP-based "taxonomy" metrics are actually measuring hierarchical understanding rather than just inheriting CLIP's well-documented inability to distinguish fine-grained semantic differences?

- If 'Specificity' ranks human-dispreferred models (like SD1.5) highly, what is its practical utility as a benchmark metric?

- If the core challenge of taxonomic generation is handling concise/abstract prompts (as suggested in Figure 1), why heavily emphasize results that use explicit definitions, which essentially convert the task back into standard descriptive T2I generation?

-  With only 4 assessors for 3370 pairs, how many human judgments were obtained per image pair?

- Why was a web-scale retrieval baseline (like LAION) not included to provide a realistic comparison for the retrieval approach?

---

> ### Author Response · Authors · 2025-11-19
> **Answer to Reviewer KcuZ**
>
> We are grateful to the reviewer for their valuable comments and concerns and are happy to respond to them and include this clarification in the paper. We invite the Reviewer to discuss weaknesses (W) in more detail:
>
> ### **W1:**
>
> We thank the reviewer for this observation, but we respectfully disagree with the conclusion that adding definitions turns the task into “standard instruction following.” In TTI settings, definitions are **not** a typical or natural form of instruction. As shown in **Figure 1**, WordNet-style glosses differ markedly from the image-oriented prompts that TTI models are trained on (e.g., captions, descriptions, conversational queries). These definitions are linguistic explanations of taxonomic concepts. Using them does not simply test instruction adherence—it tests whether the model can map a **linguistic conceptual description** to a visual referent.
> Our intention was not to improve scores via definitions, but to analyze **how different prompt formulations expose or fail to expose the model taxonomic knowledge.** The performance increase/drop with/without definitions is an informative diagnostic, revealing how models retrieve fine-grained taxonomic meaning.
>
> We have made this motivation explicit in the revised manuscript. (Section 3)
>
> ### **W2:**
>
> We thank the reviewer for the comment, but we would like to clarify that using four trained annotators (both male and female) is standard practice for controlled semantic evaluation, that ensures statistical robustness. Our setup follows common practice in prior semantic evaluation work, where 2–5 expert annotators are typical and often preferred over large, noisier crowdsourced pools (see [1,2,3] as examples).
>
> Here are more details to make the evaluation more reproducible. We will add this to the manuscript.
>
> * the recruitment platform: the ipynb published in the anonymous repo: https://anonymous.4open.science/r/TIGB-E002/labeling_example.ipynb was used
> * compensation: **free**
> * Each concept–image pair was evaluated by **one human annotator**, but each annotator covered a large portion of the dataset: **845 pairs with definitions and 845 without definitions** (1,690 judgments per annotator). Across the four annotators, this yields a substantial amount of human evaluation data.
>
> To assess consistency, we compute **Spearman correlations between annotators based on the model rankings they produce**, obtaining **$ \approx 0.8 (p \leq 0.05)$**, which indicates strong agreement. We also report the **pairwise correlations between model rankings for each annotator** to further demonstrate reliability (all $p \leq 0.05$).
>
> | _ |annotator 1| annotator 2 | annotator 3 | annotator 4 |
> |-|-|-|-|-|
> annotator 1 | - | 0.785 | 0.741  | 0.979  |
> annotator 2 | - | - | 0.745 | 0.778  |
> annotator 3 | - | -  | - | 0.750 |
> annotator 4 | - | -  | - | - |
>
> These results show that, despite one annotator per pair, the overall evaluation is stable and consistent across annotators.
>
> ### **W3:**
> **The proposed “Specificity” metric and ELO scores measure different properties.**
> Human judgements compare plain images side-by-side, which is strongly influenced by a wide variety of properties: color scheme, consistency in generation, image quality, details, etc. As well as some models are optimized for human values, as noted in line 359.
> Specificity captures the representation of the precise lemma, rather than its cohyponyms. ELO comparisons would never be able to catch this exact property.
> In fact **absence of their correlation is exactly the reason to consider this metric**, as they show uncorrelated properties. Otherwise if correlated means new metric is not showing any new information.
> Finally we note that this metric is a generalization of previously proposed idea [4], which they also found useful and well correlated with human judgements.
>
> ### **W4:**
> We appreciate the reviewer’s comment. We have improved the formatting and citations, please, refer to the updated PDF.  Regarding the Results section, our intention was to keep the primary results, such as cross-model comparisons, ranking consistency across metrics, and the main human-model correlations in the main Results section. However, additional experiments and error analysis presenting in the main text would exceed the space limitations and reduce readability. We are happy to move selected additional results into the main paper if the reviewers feel specific items would strengthen the narrative.

---

> > ### Author Response · Authors · 2025-11-19
> > **Answer to Reviewer KcuZ (part 2)**
> >
> > ### **Q1: How do you validate that your CLIP-based "taxonomy" metrics are actually measuring hierarchical understanding rather than just inheriting CLIP's well-documented inability to distinguish fine-grained semantic differences?**
> > We would like to emphasize that our proposed metrics are defined within a general probabilistic framework, as described in Appendix D. For example, Lemma Similarity is formally defined as:
> >
> > $
> > \text{Similarity}_{\text{lemma}}(v, x) := P(X = x \mid v),
> > $
> >
> > where $v$ is a concept and $x$ is an image. In this formulation, CLIP is used as a practical approximation of this conditional probability. Importantly, any embedding model with text–image alignment capabilities can be substituted into the same framework, and CLIP-like models continue to improve over time [5,6,7].
> >
> > We do not claim that the current CLIP-based metrics perfectly capture fine-grained hierarchical understanding in an absolute sense. Instead, they serve as theoretically grounded, *local* probes of how generated images relate to the WordNet structure under a fixed encoder. These metrics are used alongside human preference and reward-based evaluations, and we will revise the paper to make the scope and limitations of this approach explicitly clear.
> >
> > ### **Q2: If 'Specificity' ranks human-dispreferred models (like SD1.5) highly, what is its practical utility as a benchmark metric?**
> >
> > Specificity is not meant to replace human preference metrics, but to serve as a *diagnostic* of how precisely a model distinguishes a target concept from its co-hyponyms. Its high score for SD1.5 shows that it captures a different property: semantic focus rather than visual appeal.
> >
> > **Utilities:**
> >
> > - **Complementary signal:** Used together with ELO/reward scores to identify models that are both specific and human-preferred.
> > - **Trade-off analysis:** Highlights cases like SD1.5 (high specificity, low aesthetics) vs. FLUX/Playground (high preference, lower specificity).
> > - **Task relevance:** Beneficial in taxonomy-driven or data-curation scenarios where “correct concept vs. sibling” matters more than style.
> >
> > ### **Q3: If the core challenge of taxonomic generation is handling concise/abstract prompts (as suggested in Figure 1), why heavily emphasize results that use explicit definitions, which essentially convert the task back into standard descriptive T2I generation?**
> > Likewise in W1, we would like to specify, that in TTI settings, definitions are **not** a typical or natural form of instruction. As shown in **Figure 1**, WordNet-style glosses differ markedly from the image-oriented prompts that TTI models are trained on (e.g., captions, descriptions, conversational queries). These definitions are linguistic explanations of taxonomic concepts. Using them does not simply test instruction adherence—it tests whether the model can map a **linguistic conceptual description** to a visual referent.
> >
> > ### **Q4: With only 4 assessors for 3370 pairs, how many human judgments were obtained per image pair?**
> >
> > Each concept–image pair was evaluated by **one human annotator**, but each annotator covered a large portion of the dataset: **845 pairs with definitions and 845 without definitions** (1,690 judgments per annotator). Across the four annotators, this yields a substantial amount of human evaluation data.
> >
> > To assess consistency, we compute **Spearman correlations between annotators based on the model rankings they produce**, obtaining ** \approx 0.8 (p \leq 0.05)**, which indicates strong agreement. We also report the **pairwise correlations between model rankings for each annotator** to further demonstrate reliability (all $p \leq 0.05$).
> >
> > | _ |annotator 1| annotator 2 | annotator 3 | annotator 4 |
> > |-|-|-|-|-|
> > annotator 1 | - | 0.785 | 0.741  | 0.979  |
> > annotator 2 | - | - | 0.745 | 0.778  |
> > annotator 3 | - | -  | - | 0.750 |
> > annotator 4 | - | -  | - | - |
> >
> > These results show that, despite one annotator per pair, the overall evaluation is stable and consistent across annotators.

---

> > > ### Author Response · Authors · 2025-11-19
> > > **Answer to Reviewer KcuZ (part 3)**
> > >
> > > ### **Q5: Why was a web-scale retrieval baseline (like LAION) not included to provide a realistic comparison for the retrieval approach?**
> > >
> > > We thank the reviewer for the valuable suggestion. A web-scale retrieval baseline such as LAION is indeed valuable for comparison, and we are currently running this experiment. We use Wikimedia Commons as the main retrieval method, as it is the largest openly licensed image collection that explicitly permits redistribution and reuse under clear terms. LAION aggregates images scraped from the open web, many of which do not carry permissive licenses. To maintain a fully open and legally compliant benchmark, we therefore relied on Wikimedia Commons.
> > >
> > > We will update the paper with the results of the LAION-based retrieval experiment as soon as it completes and include a discussion of the comparison in the next version. However, currently the host https://knn5.laion.ai/knn-service is not available (as instructed in https://github.com/LAION-AI/model-retrieval/tree/main), therefore, it will take much more time and resources to host the dataset locally and retrieve from it. (We will try to do our best though)
> > >
> > > ### **All in all:**
> > >
> > > We thank the reviewer again for the valuable feedback that helps us improve the paper. Therefore, in case the reviewer thinks that our paper deserves a better chance for publication, we would be extremely grateful for improving their score, taking into consideration the answers above.
> > >
> > > ### **Bibliography**
> > >
> > > [1] Amidei et al. (2018). Rethinking the Agreement in Human Evaluation Task. COLING 2018, https://aclanthology.org/C18-1281/
> > > [2] Van der Lee, C., Gatt, A., Van Miltenburg, E., & Krahmer, E. (2021). Human evaluation of automatically generated text: Current trends and best practice guidelines. Computer Speech & Language, 67, 101151.
> > > [3] Ying Xu et al. 2022. Fantastic Questions and Where to Find Them: FairytaleQA – An Authentic Dataset for Narrative Comprehension. In Proceedings of the 60th Annual Meeting of the Association for Computational Linguistics (Volume 1: Long Papers), pages 447–460, Dublin, Ireland. Association for Computational Linguistics.
> > >
> > >
> > > [4] Baryshnikov, A., & Ryabinin, M. (2023). Hypernymy understanding evaluation of text-to-image models via wordnet hierarchy. arXiv preprint arXiv:2310.09247.
> > >
> > >
> > > [5] Zhang, B., Zhang, P., Dong, X., Zang, Y., & Wang, J. (2024, September). Long-clip: Unlocking the long-text capability of clip. In European conference on computer vision (pp. 310-325). Cham: Springer Nature Switzerland.
> > >
> > >
> > > [7] Huang, Y., Fan, Z., He, Z., Polisetty, S., Li, W., & Fung, Y. R. (2025). CultureCLIP: Empowering CLIP with Cultural Awareness through Synthetic Images and Contextualized Captions. arXiv preprint arXiv:2507.06210.
> > >
> > >
> > > [8] Yeo, Wei Jie, et al. "Debiasing CLIP: Interpreting and Correcting Bias in Attention Heads." arXiv preprint arXiv:2505.17425 (2025).

---

> > > > ### Author Response · Authors · 2025-11-25
> > > > **Invitation to discussion**
> > > >
> > > > Dear Reviewer,
> > > >
> > > > As the rebuttal deadline is quickly approaching, this is a gentle reminder to please review and respond to the authors rebuttal comments.
> > > >
> > > > If you have any concerns, clarifications, or updates to your assessment after reading the response, please share them as soon as possible so we can address them promptly before the deadline.
> > > >
> > > > If you feel that our responses adequately address the concerns raised in your initial review, we would be extremely grateful for improving the score.
> > > >
> > > > Thank you very much for your time and contributions.
> > > >
> > > > Authors.

---

### Official Review · Reviewer_tSGp · 2025-11-02

**Soundness:** 2
**Presentation:** 2
**Contribution:** 2
**Rating:** 4
**Confidence:** 3

**Summary:**

This paper introduces a new benchmark for Taxonomy Image Generation, aiming to evaluate how well text-to-image models can visually depict concepts from hierarchical lexical databases like WordNet. The authors construct datasets combining common-sense, random, and LLM-generated taxonomy concepts and evaluate 12 models using nine taxonomy-aware metrics and both human and GPT-4 pairwise feedback. Results show that model rankings on this benchmark differ markedly from traditional T2I benchmarks, with Playground-v2 and FLUX performing best overall, while retrieval-based methods perform poorly. The paper also proposes new similarity metrics grounded in KL divergence and mutual information to quantify image–concept alignment. Overall, it shows the potential of automated visual generation to extend and update structured knowledge resources like WordNet.

**Strengths:**

- The paper introduces the first comprehensive benchmark for taxonomy-based image generation, bridging a clear gap between linguistic taxonomies and visual generation evaluation.

- It combines human evaluation, GPT-4 pairwise judging, and nine theoretically grounded metrics to provide a rigorous and well-rounded assessment of model performance. It evaluates 12 diverse text-to-image models across multiple datasets, ensuring broad coverage and fair comparison of model capabilities.

- The findings reveal that taxonomy-based visual understanding differs significantly from standard T2I benchmarks, showing new challenges and directions for semantically grounded generation.

**Weaknesses:**

- The benchmark focuses mainly on WordNet concepts, which may limit generalization to other taxonomies or domains that differ in structure and it may contain outdated or linguistically biased concept definitions that could propagate into the evaluation.

- More qualitative analysis that explains why certain models fail on specific taxonomy levels (e.g., abstract vs. concrete concepts) will be helpful for improving the paper quality.

- Although human evaluation is included, the number of annotators is small, which may limit statistical robustness.

**Questions:**

How well do the proposed taxonomy-specific metrics (like hypernym and cohyponym similarity) correlate with human semantic understanding beyond CLIP-based embeddings?

---

> ### Author Response · Authors · 2025-11-19
> **Answer to Reviewer tSGp**
>
> We are grateful to the reviewer for their valuable comments and concerns and are happy to respond to them and include this clarification in the paper. We invite the Reviewer to discuss weaknesses (W) in more detail:
>
> ### **W1:**
> We thank the reviewer for pointing this out and have added this point to the **Limitations** section in Appendix A.
>
> Nevertheless, we would like to specify that our goal in this work is to analyze TTI models behavior from a **linguistic perspective**, focusing on how models represent core semantic relations such as hypernymy and co-hyponymy. WordNet is particularly well suited for this purpose: it is a carefully curated linguistic taxonomy with explicit, human-interpretable semantic relations, sense distinctions, and hierarchical structure.
>
> Moreover, WordNet has its extensive coverage (in general & in specific domains), manual curation, long-standing use in lexical semantic research, and its linkage to ImageNet.
>
> As an alternative, **Wikidata** provides images, however, its hierarchical relations (“subclass of,” “instance of”) are not linguistically curated. The resulting taxonomy is highly heterogeneous in granularity and does not support the controlled taxonomic evaluation we target.
>
> Extending the benchmark to other resources would require substantial additional design and alignment work and is therefore a **separate research direction**. In the present paper, we intentionally focus on WordNet because it provides the clearest foundation for linguistic evaluation of model representations.
>
> At the same time, our methodology and evaluation setup are resource-agnostic and can be extended to other ontologies if needed. Relational patterns are generalizable well beyond WordNet specific structure.
>
> ### **W2:**
>
> There already exists qualitative analysis on models’ failures in Appendix I. Here we briefly identify main outcomes:
> All models struggle with depicting
>
> - a. abstract concepts;
> - b. nonfrequent and specific words (”orifice.n.01” with the lemma ”rima”);
> - c. notions of people with specific functional role (”holder.n.02” with the lemma ”holder”, for example).
>
> Abstract concepts are handled with the following:
>
> 1. Text in images (and these models do not succeed in writing) in Openjourney, HDit, SD3, PixArt
> 2. Abstraction (colors, lines, etc.) in HDit, SDXL, SD3, PixArt
>
> Other unwanted behaviors for the purposes of illustrating taxonomies include:
>
> 1. Generating playing cards for the concepts (most seen in Openjourney, also present in SD1.5);
> 2. Abstract ornamental circles (also most found in Openjourne, and some in SD1.5).
> 3. Depicturing monsters when facing rare animal names (“acanthocephalan”, “gelechiid” seen in Kandinsky3).
>
> Most importantly, models struggle closer to the leaves of a taxonomy: they tend to create an image of a parent concept without necessary features of the child (PixArt for different cheese sorts, ”feeling”, “emotion”, “pain” depicted similarly in Flux).
>
> **If the reviewer was expecting to see the analysis in the main part of the paper, we would be glad to move it there.**
>
> ### **W3:**
>
> We thank the reviewer for the comment, but we would like to clarify that using four trained annotators (both male and female) is standard practice for controlled semantic evaluation, that ensures statistical robustness. Our setup follows common practice in prior semantic evaluation work, where 2–5 expert annotators are typical and often preferred over large, noisier crowdsourced pools (see [1,2,3] as examples).
>
> ### **Q1: How well do the proposed taxonomy-specific metrics (like hypernym and cohyponym similarity) correlate with human semantic understanding beyond CLIP-based embeddings?**
>
> Thank you for the question. Our taxonomy-specific metrics (hypernym and co-hyponym similarity) correlate with human semantic understanding through the **human evaluation already included in the benchmark ($ \rho \approx 0.911, p \leq 0.00004$ for Hypernym CLIP-Score, $ \rho \approx 0.871, p \leq 0.00022 $ for Co-hyponym CLIP-Score)**. This demonstrates that the proposed metrics capture relations that humans reliably recognize.
>
> Additionally, we would like to specify that **WordNet is a human-curated taxonomic structure**, which already embeds expert semantic judgments about hypernymy, hyponymy, and siblinghood. WordNet is itself the product of extensive manual linguistic annotation, and therefore provides a reliable human-grounded semantic foundation. Therefore, our evaluation benchmark already assesses how well models connect these human-defined concepts to visual evidence.
>
> We also clarified this point in the revision in Section 4.2.

---

> > ### Author Response · Authors · 2025-11-19
> > **Answer to Reviewer tSGp (part 2)**
> >
> > ### **All in all:**
> >
> > We thank the reviewer again for the valuable feedback that helps us improve the paper. Therefore, in case the reviewer thinks that our paper deserves a better chance for publication, we would be extremely grateful for improving their score, taking into consideration the answers above.
> >
> > ### **Bibliography**
> >
> > [1] Amidei et al. (2018). Rethinking the Agreement in Human Evaluation Task. COLING 2018, https://aclanthology.org/C18-1281/
> >
> > [2] Van der Lee, C., Gatt, A., Van Miltenburg, E., & Krahmer, E. (2021). Human evaluation of automatically generated text: Current trends and best practice guidelines. Computer Speech & Language, 67, 101151.
> >
> > [3] Ying Xu et al. 2022. Fantastic Questions and Where to Find Them: FairytaleQA – An Authentic Dataset for Narrative Comprehension. In Proceedings of the 60th Annual Meeting of the Association for Computational Linguistics (Volume 1: Long Papers), pages 447–460, Dublin, Ireland. Association for Computational Linguistics.

---

> > > ### Author Response · Authors · 2025-11-25
> > > **Invitation to discussion**
> > >
> > > Dear Reviewer,
> > >
> > > As the rebuttal deadline is quickly approaching, this is a gentle reminder to please review and respond to the authors rebuttal comments.
> > >
> > > If you have any concerns, clarifications, or updates to your assessment after reading the response, please share them as soon as possible so we can address them promptly before the deadline.
> > >
> > > If you feel that our responses adequately address the concerns raised in your initial review, we would be extremely grateful for improving the score.
> > >
> > > Thank you very much for your time and contributions.
> > >
> > > Authors.

---

> > > > ### Comment · Reviewer_tSGp · 2025-11-27
> > > >
> > > > Thanks for the response and I really appreciate the authors' effort. The responses addressed my concerns and I will raise my score accordingly.

---

### Meta-Review · Area_Chair_6UCf · 2026-01-05

**Summary:**

Initial reviews for this paper were negative (4 / 4 / 4 / 2). The reviewers noted that the paper provides the first analysis of image generation through text taxonomy and has some interesting methological depth with relation to the evaluation. The reviewers provided a long list of concerns:
1. Concerns about the adding the explicit definition to the prompt which make the paper more about prompt following than taxonomy (KcuZ)
2. Concerns about the generalization of taxonomic concepts (tSGp)
3. Potentially weak evaluation of models (e.g. recent 2025 models) (zHqq)
4. Sparse human evaluation section (tSGp, KcuZ)

Along with several others.

**Reviewer Concerns:**

1. Regarding the explicit definition, the authors respond that the additional definition are not part of the standard text prompts during train time and thus test whether the model can "map a linguistic conceptual description to a visual referent". In addition, the goal was to measure how different prompt formulations demonstrate taxonomic knowledge. To be honest, I'm not sure if this fully addresses the problem mentioned by the the reviewer. I think the first rebuttal point (regarding a mismatch between inference and train) may not be correct in many situations (T2I models are receiving increasingly detailed captions which describe detail elements of the scene). The second rebuttal point feels like it would be more convincing if there was other attempts to cleanly ablate this potential effect (the additional details provide more detail for the prompt which improve the generation).
2. Regarding generalization about taxonomic concepts, the authors defend WordNet and note that it would be difficult to change this.
3. Regarding the lack of recent models, the authors suggest that they will add Qwen-Image. The authors suggest that their main benefit is the benchmark. However, I disagree. The benefit of this is the combination of the benchmark and the conclusions that you can draw from the benchmark (how well models are doing). Since the conclusions are important, including recent models is also important.
4. Regarding the human evaluation, the authors add many of the details and say that 4 raters is standard.

**Reviewer Scores:**

I think some of the reviewers may have raised either scores, e.g. qiUN and tSGp who say they will. However, I do not think that the other reviewers, particularly KcuZ, would raise their scores. In general, in my opinion, there is enough outstanding concerns that were not convincingly addressed in the rebuttal that I hesitate to recommend acceptance.

---

### Decision · Program_Chairs · 2026-01-26

Reject